# Loss of Cdc13 causes genome instability by a deficiency in replication-dependent telomere capping

Rachel E. Langston[1], Dominic Palazzola[1], Erin Bonnell[2], Raymund J. Wellinger[2], Ted Weinert[1] *

1 Department of Molecular and Cellular Biology, University of Arizona, Tucson, Arizona, United States of America, 2 Department of Microbiology and Infectiology, Université de Sherbrooke, Sherbrooke, Quebec, Canada

* tweinert@email.arizona.edu

**Data Availability Statement:** All relevant data are within the manuscript and its Supporting Information files.

## Abstract

In budding yeast, Cdc13, Stn1, and Ten1 form the telomere-binding heterotrimer CST complex. Here we investigate the role of Cdc13/CST in maintaining genome stability by using a Chr VII disome system that can generate recombinants, chromosome loss, and enigmatic unstable chromosomes. In cells expressing a temperature sensitive *CDC13* allele, *cdc13^{F684S}*, unstable chromosomes frequently arise from problems in or near a telomere. We found that, when Cdc13 is defective, passage through S phase causes Exo1-dependent ssDNA and unstable chromosomes that are then the source for additional chromosome instability events (e.g. recombinants, chromosome truncations, dicentrics, and/or chromosome loss). We observed that genome instability arises from a defect in Cdc13's function during DNA replication, not Cdc13's putative post-replication telomere capping function. The molecular nature of the initial unstable chromosomes formed by a Cdc13-defect involves ssDNA and does not involve homologous recombination nor non-homologous end joining; we speculate the original unstable chromosome may be a one-ended double strand break. This system defines a link between Cdc13's function during DNA replication and genome stability in the form of unstable chromosomes, that then progress to form other chromosome changes.

## Author summary

Eukaryotic chromosomes are linear molecules with specialized end structures called telomeres. Telomeres contain both unique repetitive DNA sequences and specialized proteins that solve several biological problems by differentiating chromosomal ends from internal breaks, thus preventing chromosome instability. When telomeres are defective, the entire chromosome can become unstable and change, causing mutations and pathology (cancer, aging, etc.). Here we study how a defect in specific telomere proteins causes chromosomal rearrangements, using the model organism *Saccharomyces cerevisiae* (budding or brewer's yeast). We find that when specific telomere proteins are defective, errors in DNA

**Funding:** Funding from this work was provided by the National Institutes of Health (https://www.nih.gov) grant GM076186-5 to TW and the NIH training grant GM08659 to REL. The Canadian Institutes for Health Research (http://www.cihr-irsc.gc.ca) grant FDN154315 supported EB and RJW. The funders had no role in study design, data collection and analysis, decision to publish, or preparation of the manuscript.

**Competing interests:** The authors have declared that no competing interests exist.

replication generate a type of damage that likely involves extensive single-stranded DNA that forms inherently unstable chromosomes, subject to many subsequent instances of instability (e.g. allelic recombinants, chromosome loss, truncations, dicentrics). The telomere protein Cdc13 is part of a protein complex called CST that is conserved in most organisms including mammalian cells. The technical capacity of studies in budding yeast allow a detailed, real-time examination of how telomere defects compromise chromosome stability in a single cell cycle, generating lessons likely relevant to how human telomeres keep human chromosomes stable.

## Introduction

Maintaining chromosomes through generations of cellular divisions is an essential process. Telomeres are the highly conserved structures that cap the ends of linear chromosomes and safeguard against genome instability. Telomeres consist of repeated G-rich sequence elements (or "TG repeats") ending with a 3'-overhang [1]. One important function of telomeres is to counteract the gradual shortening of the telomeric DNA caused by conventional DNA replication, thereby solving the "end-replication problem" [2,3]. This is achieved by the telomeric TG repeats acting as a substrate for the conserved reverse transcriptase telomerase, which recognizes and extends the repeats in a highly regulated manner [4,5]. Another primary function of telomeres is to protect chromosome ends from improper recognition and processing as a DNA double-strand break (DSB) [1,6]. Intriguingly, some of the telomeric features that act to protect the chromosome end also can act as detriments to chromosome stability. For example, secondary structures formed by the G-rich repeats [7–9] and the many DNA-bound proteins can impede DNA replication through the region [10,11]. Long non-coding RNAs bound to the telomeric repeats, dubbed TERRA, can also act as a replication block [12,13] and foster recombinatorial repair [14]. In contrast to other difficult-to-replicate regions in the chromosome, DNA replication is particularly challenging in the telomere since there is no option of rescue by an oncoming replication fork if the initial replication fork should fail. General errors in DNA replication have been linked to chromosome instability, which is one of the hallmarks of cancer [15–20].

Classically, telomeric DNA binding proteins are thought to be important for maintaining the stability of the telomere itself. In *Saccharomyces cerevisiae*, one such protein is the essential single-stranded DNA binding protein Cdc13. Cdc13 has a high specificity for the terminal telomeric G-strand and can bind the G-rich ssDNA either alone or in a complex with Stn1 and Ten1 (the Cdc13/Stn1/Ten1 heterotrimer CST; [21–23]). CST has functional and structural similarity with the heterotrimeric replication protein A (RPA) and thus has also been dubbed t-RPA (telomeric RPA; [24]). Cdc13 as part of the CST complex facilitates telomere functions via several mechanisms: by facilitating telomerase-mediated telomere elongation [25,26], by assisting in telomere processing after replication [3], and by "capping" the telomere [1,27,28]. "Capping" is a term that encompasses at least two functions: a putative post-replication capping function where CST binds to the 3' ssDNA TG overhang and blocks degradation, and a DNA replication-dependent function where Cdc13 facilitates semi-conservative DNA replication through its interaction with the lagging strand machinery (even independent of telomerase extension [29,30]). CST complexes with similar functions have been identified in fission yeast, plants, and mammals [31–34]. These CST complexes have additionally been implicated in rescuing stalled replication forks in difficult-to-replicate sequences [35–38]. Mutations in hCTC1 (the functional homologue of scCdc13) and hStn1 have been linked to Coats Plus

Syndrome in humans [39,40]. While many classical telomeropathies are linked to telomerase deficiencies and short telomeres [41], the telomere dysfunction found in patients with mutated hCTC1 are consistent with errors in telomeric DNA replication [42].

In this study, we use Cdc13 and a previously developed chromosome instability assay to explore the link between uncapped telomeres and chromosome instability. Using a conditional *cdc13* mutant (*cdc13^{F684S}*) we show that defects in Cdc13 indeed lead to a high rate of instability, most prominently forming unstable chromosomes that eventually resolve to become recombinants, are lost completely, or remain and are propagated in an unstable form. Taking further advantage of the conditional *cdc13* allele, we demonstrate that cells that complete one S phase at the restrictive temperature form unstable chromosomes in G2. We also provide evidence that the critical function of Cdc13 needed to keep chromosomes stable is in its DNA replication-dependent function, and not in its putative post-replicative chromosome capping function. Furthermore, our molecular and genetic evidence suggests a key role for ssDNA in forming unstable chromosomes. Additionally, unstable chromosome formation does not require NHEJ nor HR to form, suggesting the unstable chromosome might be linear with an ssDNA "end-gap". Finally, we show telomere-linked unstable chromosomes progress to form other, more centromere-proximal rearrangements such as allelic recombinants, chromosome truncations, a specific dicentric, and chromosome loss. The results suggest a model: a Cdc13-defect disrupts replisome function, allowing ssDNA degradation and thus end-gaps on a lagging strand template, which then forms an initial unstable chromosome that progresses to other structures during subsequent cell cycles.

## Results

### Chromosome VII disome system and Cdc13-associated instability

We use the previously developed disomic chromosome system in *Saccharomyces cerevisiae* to analyze chromosome instability in *cdc13* mutants (Fig 1). This system consists of a haploid cell with an extra-numerary, and therefore nonessential, chromosome VII (Fig 1A; based on Meeks-Wagner and Hartwell 1986; [43,44]). We detect rearrangements of the extra chromosome by selecting for loss of the *CAN1* gene, located about 25 kb from the left telomere (Fig 1A). *CAN1* cells are sensitive to the drug canavanine (Can^S); when plated on canavanine plates, cells with an unchanged chromosome retain *CAN1* and die. Alternatively, cells that lose *CAN1* through any of several chromosome rearrangements are resistant to canavanine (Can^R, Fig 1B*i*) and grow on canavanine-containing media. Point mutations in *CAN1* are possible but are rarely observed in this system [45]. The disome is genetically marked to assist in determining the structure of chromosome changes. The various rearrangements lead to different growth phenotypes and marker composition (e.g. a cell that is Lys^- Leu^- Trp^- and Ade^- suffered chromosome loss; Fig 1B).

In this study, we present evidence that examines a sequence of events shown in Fig 1B*ii* [44,46]. In a first phase (Cdc13 inactivation on rich media, 30˚C), cells undergo a replication error due to Cdc13 inactivation. That cell (red and blue stripped) either undergoes repair or, upon failing to repair, forms a primary unstable chromosome (red cell). That 1˚ unstable chromosome, still in rich media, then replicates to form either another unstable chromosome, a recombinant, or chromosome loss. In the second phase, cells are allowed to resynthesize Cdc13 (shift to 25˚C), and are subjected to selection against *CAN1* (Cdc13 activation, selection). When cells from the first phase are subjected to selection, they form either a recombinant that generates a Can^R Ade^+ round colony, chromosome loss that forms a Can^R Ade^- colony, or a sectored colony. Notice that a sectored Can^R Ade^+ colony comes from a cell with an unstable chromosome (marked 2˚ unstable). Subsequently, secondary unstable

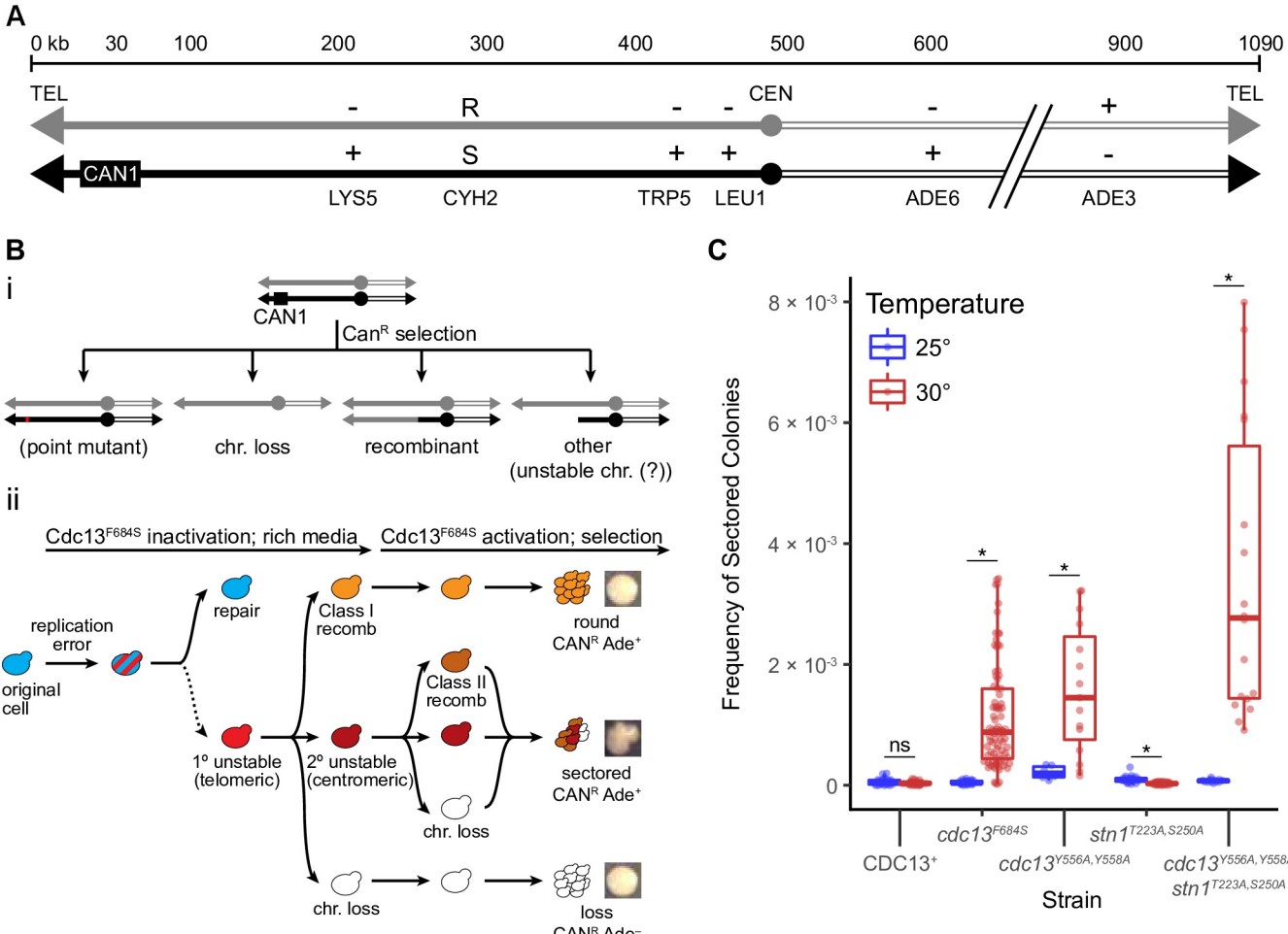

**Fig 1. Cdc13 suppresses unstable chromosomes through its activity with CST/t-RPA.** (A) Chr VII disome system. The two homologues of Chr VII are in black or grey. Position of *CAN1* and 6 heterozygous markers are shown. (B) *i*: Schematic of the different rearrangements recovered in the Chr VII assay. *ii*: Model of the unstable chromosome's fate on a cellular level. (C) Frequency of unstable chromosomes in *cdc13* mutants from cells grown at 25˚C (blue) or 30˚C (red). The median frequency, IQR, and statistically significant fold changes relative to the permissive temperature are noted (* < 0.01; Mann-Whitney U).

chromosomes in sectored colonies form a colony on selective media. As they divide, they are prone to further rearrangements and form either additional unstable chromosomes, recombinants, or chromosome loss. Note that recombinants formed in a sectored colony came from a cell with an unstable chromosome. The distinction between Class I and II recombinants is that Class I recombinants form in rich media and Class II recombinants form in selective media; thus Class I form from an unstable chromosome, and Class II form from a subsequent unstable chromosome.

We note that, in earlier studies, we speculated that instability might begin at inverted repeats located at a site 403 kb from the telomere (discussed further below; [46]). We proposed that some form of replication error in the inverted repeat region generated the first unstable chromosome. However, we have since found that deleting these inverted repeats does not alter the frequency of instability [47,48]. Therefore few, if any, of the initial unstable chromosomes form in the 403 region. Instead we determined that rearrangements at the 403 inverted repeats are secondary events from an initial event at the telomere (tested via a telomerase mutant [47]). Thus we focus on analyzing whether mutations in telomere proteins form unstable chromosomes.

## Chromosomes in *cdc13* mutants undergo genome instability at semi-permissive temperatures.

To determine the effect of a *CDC13* mutation on chromosome stability, we use a temperature-sensitive conditional mutant, $cdc13^{F684S}$ [49], as *CDC13* is an essential gene. It has been reported that a *cdc13-1* mutation generates recombinants and chromosome loss in both Chr V and Chr VII diploids (unstable chromosomes were not reported; [50,51]). We used the $cdc13^{F684S}$ allele instead of *cdc13-1* because the former is fully functional at the permissive temperature [49], whereas the commonly-used *cdc13-1* allele is compromised in our cells even at 23˚C (S7 Fig, ssDNA is present in *cdc13-1* even at permissive temperature).

We thus integrated the temperature-sensitive $cdc13^{F648S}$ allele into the Chr VII disome, grew cells at a permissive (25˚C) or semi-permissive temperature (30˚C) for ~20 generations, and plated onto selective media to detect chromosome instability (S1 Fig). The $cdc13^{F684S}$ cells generated an increased frequency of all three types of rearrangements compared to the controls (a $CDC13^+$ strain grown at 30˚C or $cdc13^{F684S}$ grown at 25˚C); namely, in the frequency of sectored colonies (Fig 1C), and in round and chromosome loss colonies (Table 1). We tested instability in one other *cdc13* temperature-sensitive mutant ($cdc13^{Y556A,Y558A}$) and found it exhibited increased instability as well (Fig 1C; Table 1). We confirmed that the sectored colonies formed in *cdc13*-defective cells contain cells of multiple phenotypes (S2 Fig), indicating that unstable chromosomes form and are detected by the sectored colony phenotype when Cdc13 is defective. While this study was in progress, we reported elsewhere that a $cdc13^{F684S}$ mutation also causes unstable chromosomes in a Chr V disome system [48]. Thus, *cdc13*-dependent unstable chromosomes are not a Chr VII specific phenomenon and can form in multiple chromosomes.

We next determined if Cdc13 prevents chromosome instability by itself or as part of the CST heterotrimeric complex with Stn1 and Ten1 [24]. We used a mutation of *STN1*, $stn1^{T223A,S250A}$, that partially disrupts Stn1's ability to concurrently bind Cdc13 and Ten1 [52]. We found that the $stn1^{T223A,S250A}$ single hypomorphic mutant showed no increase in instability (Fig 1C; Table 1). However, the double mutant $cdc13^{F684S}$ $stn1^{T223A,S250A}$ generated more sectored colonies than the $cdc13^{F684S}$ single mutant alone with a 3.2-fold increase in sectored

**Table 1. Median frequencies of chromosome instability in $cdc13^{F684S}$ at 25 and 30˚C.**

| Temp | Strain | Sectored (×10⁻⁵) | Round (×10⁻⁵) | Chromosome Loss (×10⁻⁵) |
|---|---|---|---|---|
| 25˚ | Wild Type (CDC13⁺) | 3.6 [5.6] (1.0) | 6.3 [15] (1.0) | 15 [320] (1.0) |
| | $cdc13^{F684S}$ | 3.3 [4.6] (0.92) | 5.7 [7.5] (0.90) | 130 [2000] (8.7) |
| | $cdc13^{Y556A,Y558A}$ | **20 [18] (5.6)** | **25 [11] (4.0)** | 81 [137] (5.4) |
| | $stn1^{T223A,S250A}$ | **9.5 [5.5] (2.6)** | 11 [6.7] (1.7) | 87 [130] (5.8) |
| | $cdc13^{F684S}$ $stn1^{T223A,S250A}$ | 7.7 [2.9] (2.1) | 12 [7.1] (1.9) | 110 [360] (7.3) |
| 30˚ | Wild Type (CDC13⁺) | 2.9 [3.4] (1.0) | 7.1 [4.5] (1.0) | 12 [38] (1.0) |
| | $cdc13^{F684S}$ | **88 [120] (30)** | **20 [20] (2.8)** | **73 [93] (6.1)** |
| | $cdc13^{Y556A,Y558A}$ | **150 [170] (52)** | **51 [92] (7.2)** | **210 [260] (18)** |
| | $stn1^{T223A,S250A}$ | 3.0 [2.0] (1.0) | 6.0 [7.0] (0.85) | 20 [27] (1.7) |
| | $cdc13^{F684S}$ $stn1^{T223A,S250A}$ | **280 [420] (97)** | **20 [25] (2.8)** | **140 [70] (12)** |

Median values and IQR, in []s, are reported.

In bold, statistically significant ($P < 0.01$, Mann-Whitney U) fold change between single mutants and Wild Type (CDC13⁺) or between $cdc13^{F684S}$ *mutX* and $cdc13^{F684S}$ from the corresponding temperature.

Normalized median frequencies are reported in ()s (normalized to CDC13⁺ from the corresponding temperature).

colonies compared to the $cdc13^{F684S}$ single mutant (Fig 1C). The frequency of round colonies and chromosome loss were not significantly changed from the $cdc13^{F684S}$ single mutant (Table 1). We reason that the $stn1^{T223A,S250A}$ mutation does not fully impair Cdc13's and Stn1's function at the semi-permissive temperature. As a result, CST function would be more severely defective when both $cdc13^{F684S}$ and $stn1^{T223A,S250A}$ mutations are present in the same cell, thus exacerbating the instability. We therefore conclude that the CST complex prevents chromosome instability in general, and unstable chromosomes in particular.

## Cdc13 suppresses chromosome instability at the terminal telomere repeat but not at an internal site

Next, we investigated where Cdc13 acts on the chromosome to suppress instability. In addition to Cdc13's well-explored role at telomeres, it could potentially act at internal TG-rich single-stranded sequences, for example during DNA replication [53]. We took a genetic approach, making use of the fact that the frequency of round colonies (containing Class I recombinants) is increased by a defect in $cdc13^{F684S}$ (Table 1). Since Class I recombinants likely form from the primary unstable chromosome (Fig 1B*ii*), they likely arise near the site of the initial *cdc13*-error [47,54,55]. Thus, if a telomere defect causes recombinants, we expect those recombinants to map to telomere-proximal genetic intervals (as shown previously using *cdc13-1*; [27,50]). Indeed, in our Chr VII disome system we found that, in *cdc13*-defective cells, Class I recombinants in round colonies were biased towards the telomere-proximal region compared to the probable likelihood of recombination in the region based on its size ("expected", Fig 2A). This suggests that a *cdc13*-defect causes errors in or near the telomere (as shown by Garvik *et al* 1995 for a Chr V strain; [27]). Interestingly, the events in $CDC13^+$ cells are also increased in the telomere-proximal region compared to what was expected. This suggests that instability may also originate from the telomere in wild type cells even though instability is less frequent overall.

To address if Cdc13 could suppress instability internally if there were an interstitial telomere-like sequence available, we inserted a 126 bp $TG_{1-3}$ DNA sequence at two different sites distant from the telomere. One site was 281 kb from the left telomere (Fig 2B), and another at a locus 484 kb from the left telomere (15 kb from the centromere; S3 Fig). Both sequences were inserted such that the TG repeats were oriented relative to nearby origins so that the G-rich strand was on the lagging strand template, mimicking the G-rich strand on the lagging strand template in a natural telomere (Fig 2B). The *URA3* gene was inserted into the same regions of the $cdc13^{F684S}$ strain without linked TG repeats as a control. These strains were then analyzed for instability, focusing first on the $^{281}$TG site in $cdc13^{F684S}$ mutants. We found that the internal TG repeats did not significantly change the frequencies of chromosome instabilities. If instability initiates at this site, the effect is subtle compared to instability elsewhere on the chromosome (S4 Fig). To identify any subtle effects by the internal TG repeats, we analyzed the regions of recombination in round colonies, and found that Class I rearrangements in round colonies were still biased to the telomere-proximal region (in both experimental *URA3:TG* or in the *URA3* control strains; Fig 2C); indicating that the internal TG sequence at the 281 kb site does not detectably induce events. We made similar observations for the TG repeats inserted 484 kb from the left telomere; there were no changes in the frequency of instability nor in the distribution of recombinants (S3 Fig). We therefore conclude that chromosome instability in $cdc13^{F684S}$ mutants in our system originates only from the telomere end, and not from internal TG repeats of 126 bp. It is possible that the *cdc13*-induced error could still occur at the interstitial TG repeats, only to be rescued by the oncoming replication fork before it could fully develop into an unstable chromosome. This would not be an option at the

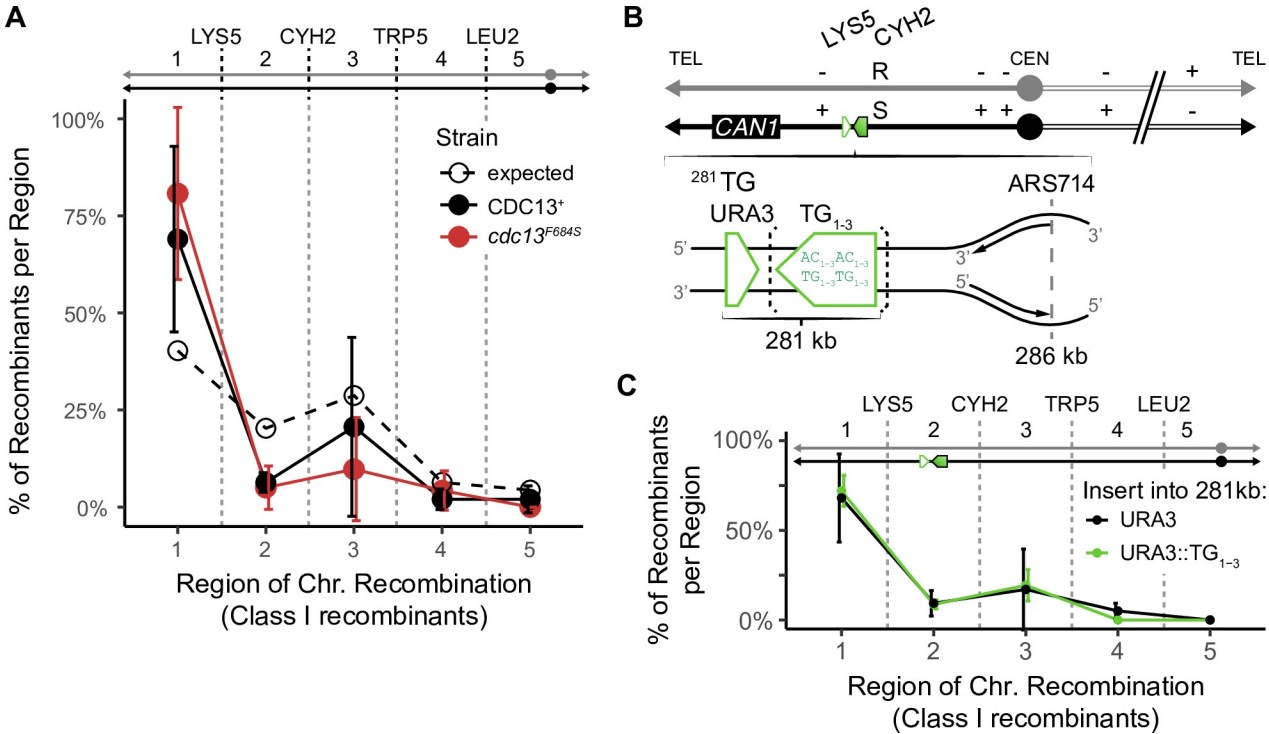

**Fig 2. Unstable chromosomes in *cdc13*<sup>F684S</sup> form at the telomere, not at internal TG repeats at a semi-restrictive temperature (30°C).** (A) Distribution of Class I recombinants from *cdc13*<sup>F684S</sup> round colonies (solid) and the expected distribution (dashed; calculated from the proportion of the region length vs total length of the chromosome arm). The average percentage and standard deviation for 4 independent experiments are shown ($P = 0.0349$; one sample *t* test). (B) Schematic for insertion of the TG$_{1-3}$ repeat (green box) into the left arm of Chr VII. The TG$_{1-3}$ insert (green box; bracketed) was marked with *URA3* (green triangle) and was inserted 281 kb from the telomere end. As a control, *URA3* alone was inserted into the same region. (C) Distribution of Class I recombinants from round colonies from *cdc13*<sup>F684S</sup> with *URA3* or *URA3*::TG$_{1-3}$ inserted into the 281 kb locus. The average percentage and standard deviation for 3 independent experiments are shown.

native telomere. It is important to note that the 126 bp interstitial telomeric repeat still influences the progression of instability even if instability initiates elsewhere.

## Cells traversing one cell cycle with inactive Cdc13 have high instability

To examine molecular events that underlie instability, we asked if inactivation of Cdc13 in a single cell cycle might generate loss of *CAN1* to form Can<sup>R</sup> cells. The data reported thus far, in Fig 1C and Table 1, were collected by growing *cdc13* mutant cells at a semi-permissive temperature (where cells delay and resume cell divisions repeatedly for about 20 generations) before being subjected to selection. Thus, we do not know when during these 20 generations events occur, and certainly cannot specify a particular cell cycle phase.

We therefore asked if we could detect instability arising in one cell cycle limited for Cdc13 function. We synchronized *cdc13*<sup>F684S</sup> cells grown at the permissive temperature (25°C) in early S phase with hydroxyurea (HU); HU-arrested cells have mostly unreplicated and stable chromosomes (Fig 3). We then shifted the HU-synchronized cells to the restrictive temperature (25°C to 37°C), washed out the hydroxyurea, and allowed cells to traverse one cell cycle at the restrictive temperature (referred to as the "+HU 25°C, -HU 37°C" experiment; Fig 3A). Western blotting verified that the Cdc13<sup>F684S</sup> protein is quickly degraded when cells are incubated at 37°C (Fig 3B). Cells completed one S phase and arrested at the *RAD9* G2/M checkpoint (Fig 3C; [56]). We then allowed resynthesis of Cdc13<sup>F684S</sup> by shifting to the permissive temperature

(25˚C), allowing cells to repair their chromosomes and resume cell division. We plated cells on selective media to determine if any of these cells had generated altered chromosomes.

The results from this HU and temperature shift experiment were surprising. Cells subjected to HU and then shifted up to 37˚C generated a very high frequency of unstable chromosomes (103-fold increase after 4 hours at 37˚C; Fig 3D). Cells subjected to HU-arrest alone were stable: there was little to no increase in all three types of chromosome changes (+HU, 25˚C; Fig 3D blue). Notably, we also found that while the temperature shift did increase the frequency of unstable chromosomes, this regime did not appreciably increase the frequency for either round colonies (containing Class I recombinants) or chromosome loss; we interpret this

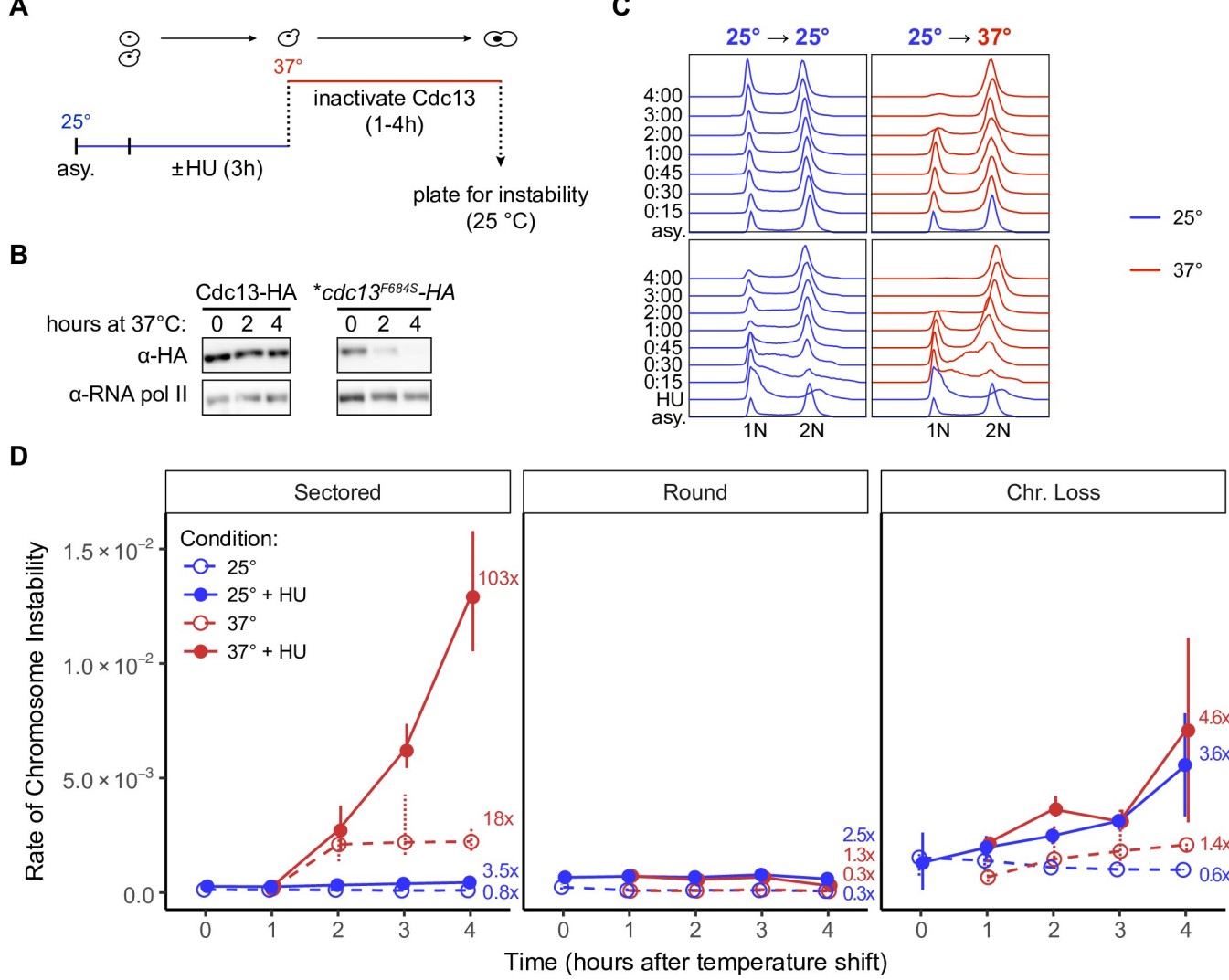

**Fig 3. Unstable chromosomes in *cdc13^F684S* form in a single cell cycle.** (A) The experimental protocol for generating unstable chromosomes in a cell cycle. Asynchronous cells were either shifted directly to 37˚C or exposed to HU (0.2 M) for 3h at 25˚C before the HU was washed out and the cells were shifted to 37˚C for 1–4 hours, then plated for instability at 25˚C. (B) Exponentially growing Cdc13-HA or *cdc13^F684S*-HA cells were shifted from 25˚C to 37˚C for 2 or 4h. Whole cell extracts were prepared and analyzed by western blot with an anti-HA antibody. Anti-RNA pol II was used for a loading control. *cdc13^F684S*-HA also contained the *cdc15-2* mutation, further explained in Fig 4. (C) FACS analysis of DNA content. Cells were grown in liquid YPD and prepared as in 2A. After HU arrest 70% of cells were in G1/S phase. The time increments refer to time after HU wash and shift to either 37˚C or 25˚C. (D) The frequency of sectored, round, and chr. loss colonies in *cdc13^F684S* from 0 to 4 hours after the temperature shift. Blue: incubated at 25˚C; red: incubated at 37˚C. Fold change and significance are calculated from the 0 hour frequency. Data shown are the medians and IQR of 4 independent experiments.

finding below. Finally, shifting to 37˚C without presynchrony in HU increased sectored colonies by 18-fold whereas with presynchrony in HU events were increased 108-fold (and again, only sectored colonies were increased, not round or chromosome loss); therefore there appears to be synergy between Cdc13 inactivation and HU treatment.

All together, these results suggest the following. Upon inactivation of Cdc13 and progression through one cell cycle, cells arrest in G2 with damaged chromosomes. The damaged chromosomes usually repair faithfully (>99% of viable cells are still Can$^S$), yet about 1 in 100 cells develop an unstable Chr VII. Surprisingly only sectored colonies are formed (each from a cell with an unstable chromosome), but neither round colonies nor chromosome loss are observed. We therefore conclude that Class I recombinants and chromosome loss are not generated directly from a Cdc13 defect, but rather arise only from unstable chromosomes (Fig 1B, extended model in S5 Fig). These results also indicate an interaction between HU-induced DNA replication stalls and Cdc13's role at the telomere during S phase. In conclusion, this experimental paradigm allows us to conclude that unstable chromosomes form from a telomere lesion first, and those then generate chromosome loss and recombinants. This paradigm should allow us to narrow down the timeframe for when Cdc13 must be active, in S phase and/or in G2 phase, to preserve chromosome stability.

## Cdc13 acts in S phase rather than in G2 to prevent instability

Cdc13 may prevent chromosome instability by any of four mechanisms: facilitating telomerase recruitment, facilitating C-strand fill-in, capping the chromosome end (specifically post-replication capping), and/or by facilitating semiconservative DNA replication. It is unlikely that Cdc13 must facilitate telomerase activity during one cell cycle to keep chromosomes stable, as a telomerase defect takes many cycles to manifest instability [47,55]. We therefore suggest that Cdc13's role in chromosome stability comes from its role in lagging strand synthesis in S phase or in chromosome capping in G2/M.

To distinguish between these two roles, we modified the single cell cycle assay described above. We performed two experiments (Fig 4A). In the first, Cdc13 is inactive in both S and in G2/M phases. In the second, Cdc13 is active in S phase and inactive in G2/M phase. To inactivate Cdc13 in both S and G2/M phases, we repeated the experiment reported in Fig 3. Asynchronously growing cells were shifted from 25˚C to 37˚C for four hours, during which they complete one cell division and arrest in G2/M; Cdc13 is therefore inactive in both S and G2/M phases. The G2/M-arrested cells were then returned to the permissive temperature and plated for instability. As expected from Fig 3D, we detected an increase in unstable chromosomes and not an increase in round colonies or chromosome loss (Fig 4D).

The second, more elaborate experiment determines the fate of chromosomes in cells where Cdc13 was active in S phase and inactive in G2/M phase. We grew *cdc13$^{F684S}$* cells at 25˚C, allowed them to progress normally through S phase, and then arrested them in G2/M at 25˚C with the spindle poison nocodazole. We then shifted the G2/M arrested cells to the restrictive temperature (37˚C) for four hours, while keeping the cells in nocodazole; thus, Cdc13 is inactive in G2/M. We confirmed that the Cdc13$^{F684S}$ protein is equally unstable at 37˚C in both treatments (Fig 4B). However, in preliminary experiments we found that about 20% of cells escaped nocodazole arrest during the four hours at high temperature (verified by flow cytometry and nuclear staining (DAPI); Figs 4C and S6). We therefore introduced a *cdc15-2* mutation that causes an arrest in telophase to prevent these escaping cells from progressing through to the next cell cycle; S6 Fig [57,58]. We then shifted the G2/M arrested cells from the restrictive to the permissive temperature to resynthesize Cdc13$^{F684S}$, washed out the nocodazole, and plated cells for instability. Remarkably, there was no increase in the frequency of sectored or

round colonies, or chromosome loss (G2/M; Fig 4D). To summarize, we found that inactivation of cdc13$^{F684S}$ in S phase causes instability, but inactivation in G2/M does not. We therefore conclude that it is Cdc13's function during DNA replication that is critical for chromosome stability, either by assisting semi-conservative DNA replication or through its role in C-strand fill-in. Either way, it's Cdc13's role during S phase that protects against chromosome instability, not its role in post-replication chromosome capping.

We also implemented this cdc13-inactivation in a version of the "G2/M-only" strategy, using arrest at the cdc15-2 telophase arrest point. We grew cells at 25˚C, treated them with nocodazole at 25˚C, then shifted the cells to the restrictive temperature after washing out nocodazole, whereupon cells exit metaphase and arrest in telophase. After 4 hours at 37˚C, telophase-arrested cells were shifted to 25˚C and assessed for instability. Again, we found that inactivating cdc13 in G2/M and in telophase did not cause an increase in instability ("T" arrest; S6 Fig).

## Inactivation of cdc13$^{F684S}$ in S phase, but not in G2/M, causes extensive ssDNA in telomeres

A Cdc13-defect is known to generate telomere-proximal ssDNA, wherein the 5' strand is degraded, leaving the 3' TG rich strand unpaired. A plethora of studies have examined this so-called "chromosome capping" role of Cdc13 [25,27,59–63]. However, it remains unclear whether the ssDNA arises due to a defect in DNA replication as cells progress through S phase, or a defect in post-replication chromosome capping. As instability arises only when Cdc13 is defective during S phase, we wished to determine if ssDNA arises only during S phase when Cdc13 is defective as well. We therefore measured ssDNA in telomeres when Cdc13$^{F684S}$ was defective in S and G2/M phase, versus defective in G2/M only, using the well-established native in-gel hybridization detection method ([63,64]; see Methods). We detected extensive ssDNA when Cdc13 was inactive in S and G2/M phase, and very little ssDNA when Cdc13 was inactive in G2/M alone (Fig 4E). Thus, generation of ssDNA occurs primarily as cells transit S phase when Cdc13 is defective, and formation of ssDNA correlates with formation of unstable chromosomes. We note that four studies have detected ssDNA in G2/M arrested cells, though only the first two cited compared the amount of ssDNA generated during S phase versus in G2/M [62,63,65,66]. Comparative studies also used nocodazole to limit cell cycle progression at the high temperatures. We believe that nocodazole at 37˚C is too leaky to maintain an efficient arrest. Indeed, we find that cdc13$^{F684S}$ CDC15$^+$ cells shifted to 37˚C with nocodazole generated ssDNA, most likely due to some cells escaping the nocodazole arrest; S7 Fig).

## An unexpected phenotypic lag of Can$^S$ probably enables the single cell cycle study

Before we did the previous experiments, there were three features of our Chr VII disome system that suggested that we would in fact not detect loss of CAN1 gene in one cell cycle even if it were to occur. First, the CAN1 gene is 25 kb from the telomere; a telomere defect due to Cdc13 dysfunction must inactivate the CAN1 gene many kilobases away from the telomere within several hours. Second, the gene may only be gone from one of the two sister chromosomes; a G2-arrested cell may have one CAN1 gene still intact, while its sister may have lost its CAN1 gene. And finally, even in the absence of the CAN1 gene, a phenotypic lag may persist until the Can1 protein is cleared from the cell surface. The appearance of Can$^R$ cells within one cycle can be explained by the phenotypic lag of canavanine. Specifically, we found that Can$^S$ cells on canavanine remain alive for several hours and sustain 1 or 2 slow cell divisions (S8 Fig). We surmise that while the uptake of canavanine is eventually crippling to a

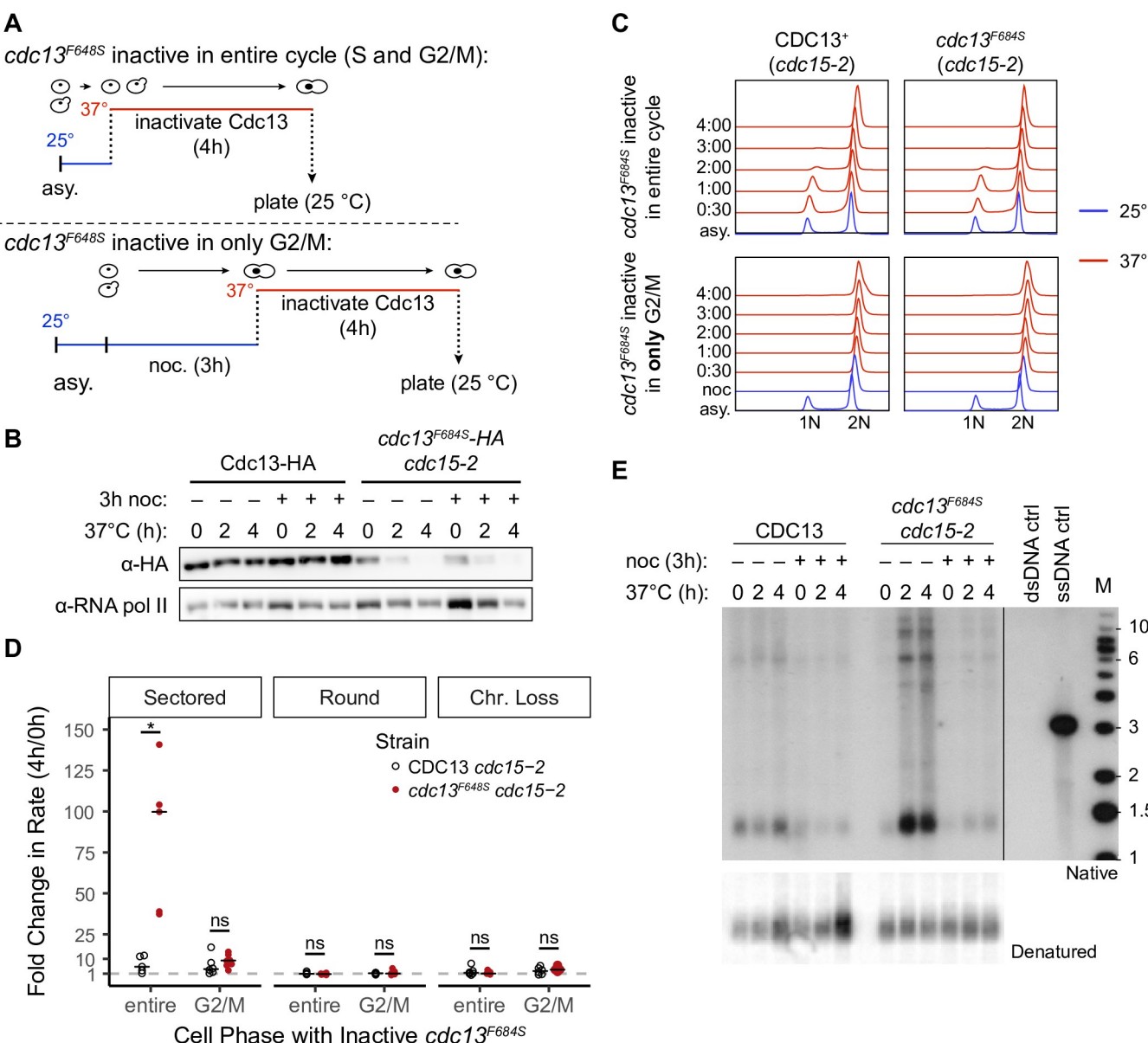

**Fig 4. Cdc13 suppresses the formation of unstable chromosomes during S phase.** (A) The experimental system to either inactivate or activate *cdc13^F684S* during S phase. *Top*: inactive *cdc13^F684S* for the entire cell cycle, cells were grown to mid-log then incubated at 37˚C for 4 hours. Aliquots for the instability assay were taken at 0h (before the temp. shift) and after 4 hours of incubation at 37˚C. Bottom: inactivate *cdc13^F684S* in only G2/M, cells were grown to mid-log at 25˚C and then arrested with nocodazole (noc). After noc was washed out, cells were shifted to 37˚C and kept arrested in G2/M with additional noc. Aliquots for the instability assay were taken at 0h (before the temp. shift) and after 4 hours of incubation at 37˚C. (B) Exponentially growing Cdc13-HA or *cdc13^F684S-HA cdc15-2* cells were shifted from 25˚C to 37˚C for 2 or 4h. Whole cell extracts were prepared and analyzed by western blot with an anti-HA antibody. Anti-RNA polII was used for a loading control. The–nocodazole (noc) samples were shown in 3B and are shown here for comparison. (C) FACS analysis of DNA content. Cells were grown in liquid YPD as depicted in Fig 4A, and cells were taken at the indicated time points for FACS analysis. Blue traces: 25˚C, red traces: 37˚C. After noc arrest 99% of cells were in G2/M. The time increments refer to time after noc removed and the temperature was shifted. (D) The median fold change in frequency of unstable chromosomes between 4h and 0h (before the temp. shift). The relevant genotypes are as follows: open circles: CDC13+ control (*cdc15-2*); closed circles: *cdc13^F684S* (*cdc13^F684S cdc15-2*). The fold changes were calculated by dividing each 4h frequency by the corresponding 0h frequency for that particular sample (n = 5). The median was then calculated (horizontal bar). The statistical significance between strains per experiment is shown (* < 0.05; Mann-Whitney U). (E) Non-denaturing in-gel hybridization using a CA oligonucleotide probe with XhoI-digested DNA from the indicated samples (*top*). To control DNA input, the gel was denatured and transferred to a nitrocellulose membrane and hybridized with a telomeric probe (*bottom*).

CAN1$^+$ cell, cells do not die immediately (S8 Fig). We therefore speculate that the lag of cell death allows cells to divide, to thus form one *CAN1*$^+$ and one *can1*$^-$ cell, during which the Can1 protein can be cleared from the *can1*$^-$ cell. Whatever the exact mechanism of *CAN1* inactivation, lack of Cdc13 causes chromosome damage in the first cycle that manifests into a Can$^R$ cell.

## Genetic analyses also suggest that ssDNA causes instability

We next used genetics to gain further insight into how unstable chromosomes are formed. We measured the frequency of instability in a series of double mutants (*cdc13*$^{F684S}$ *mutX*) to identify possible synergistic, suppressive, or epistatic interactions. We grew single and double mutants at the semi-permissive temperature (30˚C) for about 20 generations, then determined instability by plating on selective media. We found that three mutations suppressed the formation of unstable chromosomes (sectored colonies), including mutations in the nuclease Exo1 and in the helicase Pif1 (Table 2, S1 Table). Suppression by *exo1Δ* was the most robust. Curiously, while an *exo1Δ* mutation led to a decrease in sectored colonies, it led to a concomitant increase in round colonies, which consist of Class I recombinants. We infer that ssDNA formed during an S phase defect is important for instability, and when limited by an *exo1Δ* mutation, the initial lesion is converted into a Class I recombinant rather than persisting as an unstable chromosome.

We also tested for a role for the endonuclease, Sae2, known to cleave hairpins and degrade 5'-3' from a double strand break [67–69]. At the telomere Sae2 prevents the accumulation of ssDNA [70], so a *cdc13*$^{F684S}$ *sae2Δ* mutant should have increased ssDNA. Thus we predicted that instability would increase in *cdc13*$^{F684S}$ *sae2Δ*. Instead we found that *cdc13*$^{F684S}$ *sae2Δ* mutants had a *lower* frequency of instability than *cdc13*$^{F684S}$ (Table 2).

**Table 2. Median frequencies of chromosome instability in *cdc13*$^{F684S}$ mutants at 30˚C.**

| | Strain | Sectored ($\times 10^{-5}$) | Round ($\times 10^{-5}$) | Chr. Loss ($\times 10^{-5}$) |
|---|---|---|---|---|
| | Wild Type (CDC13$^+$) | 2.9 [3.4] (1.0) | 7.1 [4.5] (1.0) | 12 [38] (1.0) |
| | *cdc13*$^{F684S}$ | **88 [120] (30)** | **20 [20] (2.8)** | **73 [93] (6.1)** |
| Nucleases | *exo1Δ* | **11 [13] (3.8)** | **48 [15] (6.8)** | 44 [100] (3.7) |
| | *cdc13*$^{F684S}$ *exo1Δ* | **5.7 [3.8] (2.0)** | **380 [69] (54)** | **20 [21] (1.7)** |
| | *sae2Δ* | **7.3 [2.6] (2.5)** | 10 [7.7] (1.4) | 32 [33] (2.7) |
| | *cdc13*$^{F684S}$ *sae2Δ* | **35 [16] (12)** | 8.9 [5.5] (1.3) | 89 [55] (7.4) |
| RPA mutants | *rfa1-t33* | **260 [96] (90)** | 20 [13] (2.8) | **1300 [450] (110)** |
| | *cdc13*$^{F684S}$ *rfa1-t33* | **270 [55] (93)** | 10 [4.6] (1.4) | **870 [390] (73)** |
| | *rfa1-t33 sae2Δ* | **130 [17] (45)** | **3.9 [3.8] (0.55)** | **910 [260] (76)** |
| HR | *rad52Δ* | 16 [4.7] (5.5) | < 1 | 14 [11] (1.2) |
| | *cdc13*$^{F684S}$ *rad52Δ* | 230 [320] (79) | < 1 | **1200 [1500] (100)** |
| NHEJ | *lig4Δ* | 2.8 [2.1] (1.0) | 8.6 [6.9] (1.2) | 12 [13] (1.0) |
| | *cdc13*$^{F684S}$ *lig4Δ* | 74 [120] (26) | 16 [30] (2.3) | **12 [31] (1.0)** |
| DNA damage checkpoint | *rad9Δ* | **54 [35] (19)** | 8.0 [4.6] (1.1) | 35 [130] (2.9) |
| | *sae2Δ rad9Δ* | 87 [52] (30) | **28 [16] (3.9)** | **290 [290] (24)** |
| | *cdc13*$^{F684S}$ *rad9Δ* | **280 [330] (97)** | 12 [8.2] (1.7) | **960 [1100] (80)** |

Median values and IQR, in []s, are reported.

In bold, statistically significant ($P < 0.01$; Mann-Whitney U) fold change between single mutants and Wild Type (CDC13$^+$) or between *cdc13*$^{F684S}$ *mutX* and *cdc13*$^{F684S}$ (except for *rfa1-t33 sae2Δ* and *sae2Δ rad9Δ*, which are in relation to *rfa1-t33* and *rad9Δ* respectively).

Normalized median frequencies are reported in ()s (normalized to CDC13$^+$).

Sae2 has also been shown to antagonize Rad9 accumulation at DSBs [71]. Perhaps removing Sae2 rescued chromosome instability in *cdc13*$^{F684S}$ by allowing for additional Rad9 binding. In fact, we found that the frequency of sectoring in a *sae2Δ rad9Δ* double mutant was similar to a *rad9Δ* single mutant; *sae2Δ* did not suppress instability in a *rad9Δ* mutant. We were unable to make *cdc13*$^{F684S}$ *sae2Δ rad9Δ* triple mutants (we did recover triple mutants but they had peculiar properties requiring further study).

Finally, the rescue of *cdc13*$^{F684S}$ instability by *sae2Δ* (in a *RAD9*$^{+}$ cell) eliminated another appealing model for Chr VII instability based on a proposal by Deng *et al*. They showed that a *sae2* mutation enhanced instability in a *rpa1-t33* mutant in a Chr V GCR assay [72]. Their interpretation of *sae2Δ*'s enhancement was that extensive ssDNA of a *rpa1-t33* mutant lead to a foldback hairpin structure that, when not cleaved by Sae2, lead to instability via replication of the uncleaved hairpin and the subsequent formation of a dicentric chromosome [72]. Again we found the *sae2Δ* rescued *cdc13*$^{F684S}$-dependent instability. We also asked if Sae2 might suppress *rpa1-t33*-dependent instability in our Chr VII disome system, as Sae2 suppresses *rpa1-t33*-dependent instability in the Deng *et al* Chr V system [72]. We found that *rpa1-t33* mutants had a high frequency of instability in our Chr VII (as in their Chr V system), but, in contrast to Deng *et al*, we found that Sae2 enhanced the instability of *rpa1-t33* in our Chr VII system whereas Sae2 suppressed instability in their Chr V system (in other words *rpa1-t33 sae2Δ* had a *lower* level of instability than *rpa1-t33 SAE2*$^{+}$ (Table 2; [72]). In conclusion, the mechanism described by Deng *et al* is not responsible for the formation of unstable chromosomes in our system.

## NHEJ and HR are not involved in the initiation of instability, and the RAD9 checkpoint suppresses instability

We generated numerous other *cdc13*$^{F684S}$ *mutX* double mutants to probe mechanisms of how unstable chromosomes are formed. We tested the involvement of NHEJ and HR, motivated by the idea that sisters might need to fuse to form the initial unstable chromosomes. We found that neither *cdc13*$^{F684S}$ *lig4Δ* nor *cdc13*$^{F684S}$ *rad52Δ* double mutants had altered frequencies of sectoring relative to the corresponding single mutants (Table 2), therefore neither Lig4 nor Rad52 are needed to form unstable chromosomes (as we had concluded in an earlier study of instability arising spontaneously; see [44,46]). We did find that round colonies are dramatically reduced in *cdc13*$^{F684S}$ *rad52Δ* double mutants, as one might expect since round colonies are made up of Class I recombinants and HR is involved in recombination between homologs. We are unclear by what pathway *cdc13*$^{F684S}$ *rad52Δ* double mutants form Class II recombinants.

We additionally tested many other mutations seeking insight into mechanisms. The only mutants with greater than 2-fold increase in instability were *tel1Δ* and *rad9Δ*; the *cdc13* and *tel1Δ* interaction appears to be additive, while the *cdc13* and *rad9Δ* interaction is more synergistic (a 5-fold increase in instability over either single mutant; Table 2, S1 Table). Cdc13 and Tel1 may therefore act independently, while Rad9 may suppress instability by either limiting degradation by suppressing Exo1 [73,74] by suppressing cell division of cells with an unstable chromosome, or by both mechanisms. Notably, Cdc13 and Rfa1 appear epistatic, indicating that they are in the same pathway when it comes to protecting the cell from instability arising in the telomere. Interestingly, *rfa1-t33* instability in the Chr VII disome may arise mostly from telomere defects.

## Progression of telomere-proximal unstable chromosomes

Finally, we wished to gain further insights into early events by characterizing the fates of the telomere-proximal unstable chromosome. We have already shown that telomere-proximal unstable chromosomes in a *cdc13*$^{F684S}$ mutant generate Class I recombinants that are enriched

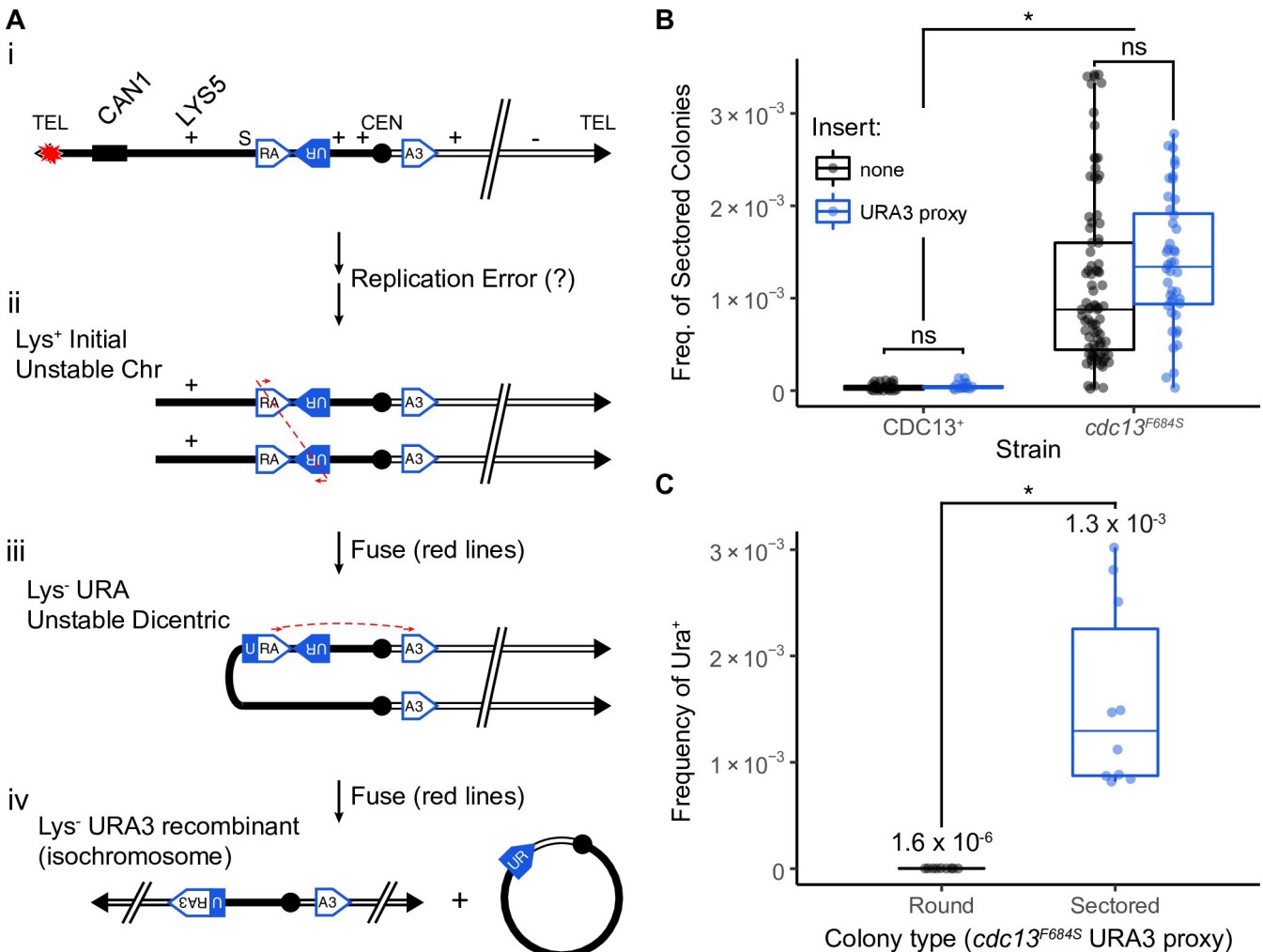

**Fig 5. *cdc13^F684S*-induced unstable chromosomes progress to centromere-proximal dicentric at a semi-restrictive temperature (30˚C).** (A) Model for the URA3 proxy translocation. Two modules, RA and RU are inserted into the left arm as inverted repeats, and the third module (A3) is inserted into the right arm (i). An inverted fusion between U<u>R</u> and <u>R</u>A forms a dicentric chromosome (ii). A secondary event, a direct repeat fusion between UR<u>A</u> and <u>A</u>3, (iii) removes a centromere and (iv) creates a functional *URA3* gene. (B) Frequency of sectored colonies between strains with (blue) and without (black) the *URA3* proxy system is unchanged. Fold change and statistical significance relative from the corresponding Wild Type frequency are noted (* < 0.01; Mann-Whitney U). (C) Frequency of *URA3^+* cells in round (black) and sectored colonies (blue) in *cdc13^F684S*. The statistical significance between round and sectored colonies is depicted (* < 0.01; Mann-Whitney U).

near the telomere, as well as chromosome loss (Fig 2A, Table 1). We next tested if the telomere-proximal unstable chromosomes also progress to form the previously characterized dicentric at the 403 kb locus [44,46]. To detect this dicentric we employed a genetic strategy developed previously called the "*URA3* proxy system" (Fig 5A; [46]). In brief, in a normal Chr VII an inverted repeat centered 403 kb from the telomere undergoes two consecutive fusion reactions, the first one between inverted repeats to form a dicentric and the second between direct repeats to resolve the dicentric to a monocentric isochromosome. The *URA3* proxy system recapitulates these two reactions, forming a dicentric and then a monocentric isochromosome, but using DNA fragments generated from the *URA3* gene such that completion of the two reactions generates an intact *URA3* gene in what was a *ura3^-* cell, and thus generating a Ura^+ cell (Fig 5A). We generated a *cdc13^F684S* mutant with the appropriate *URA3* fragments

(which did not alter the frequencies of instability; Fig 5B), and analyzed sectored colonies formed at 30°C. Indeed, Ura$^+$ cells arose about 1,000 fold more frequently in sectored colonies (founded by a telomere-proximal unstable chromosome) than in controls (round colonies founded by a stable Class I recombinant; Fig 5C). We conclude that a telomere-proximal unstable chromosome can progress to form a centromere-proximal dicentric, which resolves to an isochromosome. In addition, we think it unlikely that dicentrics initiate directly at the 403 site because we found evidence of telomere-proximal rearrangements in sectored colonies. Specifically we found both Lys$^+$ cells and Ura$^+$ cells in each of 10 sectored colonies analyzed. This indicates that telomere-proximal unstable chromosomes in *cdc13* mutants progress to form Lys$^+$ recombinants as well as the 403 kb dicentric, and subsequently the isochromosome.

## Class I and Class II recombinants

In studying unstable chromosome progression, we discovered a novel rearrangement present in strains that contain the 126 base pair TG repeat. Specifically, when the 126 base pair TG repeat (see Fig 2B) was inserted into a site 281 kb from the telomere, we found no increase in Class I recombinants linked to this site, but we did detect a dramatic increase of Class II recombinants linked to this site (Fig 6). We analyzed these Class II recombinants by pulse field gels and found that most contained a chromosome truncation corresponding in length to truncation at the inserted TG repeat (S9 Fig). As expected, we did not detect an increase in Class II recombinants when we inserted the TG repeat in the opposite orientation, where it could not act as a seed for *de novo* telomere addition (S10 Fig). We conclude that a *cdc13* defect in the telomere leads initially to an unstable chromosome that can resolve both to Class I recombinants near the telomere, and to persistent unstable chromosomes that later resolve to form truncations hundreds of kilobases from the telomere end (Class II recombinant). The insertion of the TG repeats allowed for these truncations to be stabilized.

The differing profiles of Class I and Class II recombinants is not unique to the $^{281}$TG containing strains. In some strains Class I and Class II recombinants have similar profiles (Fig 6B; *cdc13$^{F684S}$*), but in other strains the profiles differ dramatically (*rfa1-t33*, Fig 6B; *cdc13$^{F684S}$* $^{281}$TG, Fig 6C). We suggest that, in *cdc13* mutants alone, the primary and secondary unstable chromosomes do not progress extensively before they undergo recombination to form allelic recombinants. Consequently Class I and II recombinants have similar profiles. In contrast, in *cdc13$^{F684S}$* $^{281}$TG strains the progression from early to later unstable chromosomes is either more rapid or the additional TG repeats allow for recombination between the telomere and the $^{281}$TG site. As a consequence, more unstable chromosomes resolve at the $^{281}$TG site and the overall profile shifts compared to the $^{281}$TG$^-$ profile. Finally, *rpa1-t33* mutants also exhibit different Class I and Class II profiles; perhaps progression is more rapid for unstable chromosomes in an *rfa1* mutant (different profiles) than in a *cdc13* mutant (similar profiles) on selective media. Whatever the mechanisms, the distinction between Class I and Class II recombinants suggests the unstable chromosomes are very dynamic.

## Discussion

Here we report how chromosome instability is caused by a defect in the telomere-binding protein Cdc13, acting in the CST complex. First, we showed that instability is due to a defect in Cdc13/CST, and instability is linked to the TG repeats at the *bonafide* telomere and not to TG repeats inserted internally (Figs 1 and 2). Events linked to the internal TG repeats do arise, but we find they arise from a previously formed telomere-linked unstable chromosome as Class II recombinants (Fig 6). Second, the unstable chromosomes originate from damaged chromosomes containing ssDNA (Fig 4), and the ssDNA is generated from a mutant Cdc13-associated S phase defect

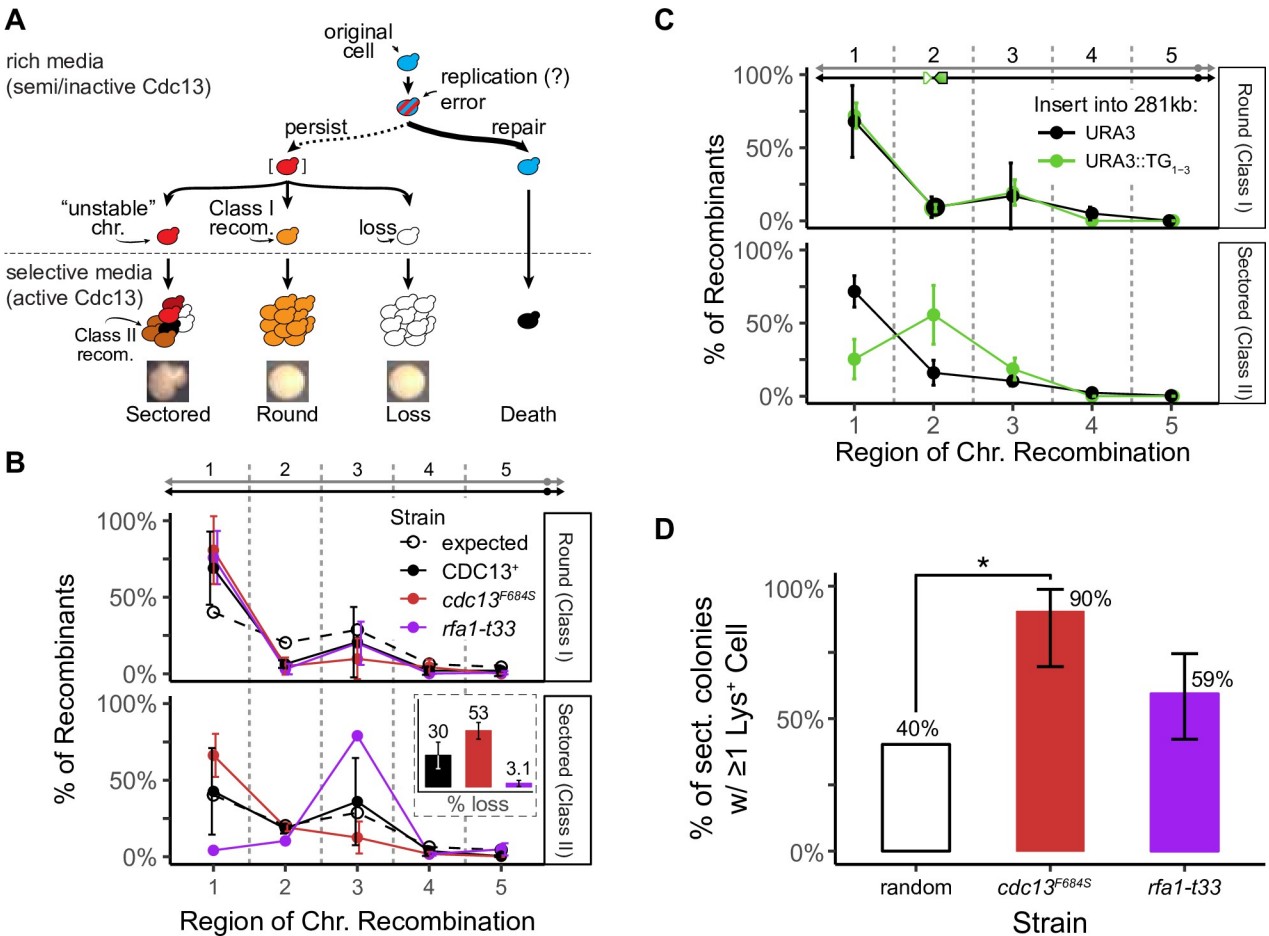

**Fig 6. Class I and Class II recombinants at a semi-restrictive temperature (30˚C).** (A) Schematic of how diversity arises from one unstable chromosome. (B) Distribution of genotypes from round (Class I recombinants) and sectored colonies (Class II recombinants) from *cdc13^F684S* (black) or *rfa1-t33* (purple). The expected distribution ("random") is dashed. The average percentage and standard deviation for 3 independent experiments are shown. *Inset*: the frequency of chromosome loss within the sectored colony for *cdc13^F684S* (black) and *rfa1-t33* (purple). (C) Distribution of Class II recombinants from sectored colonies from *cdc13^F684S* with *URA3* or *URA3*::TG_{1-3} inserted into the 281 kb locus. The average percentage and standard deviation for 3 independent experiments are shown. (D) Average percentage with 95% confidence interval of sectored colonies that retained at least 1 Lys^+ cell for *cdc13^F684S* and *rfa1-t33*. The expected distribution ("random") is white (* < 0.01, one sample *t* test).

and degradation by Exo1 (Table 1). In addition, instability is not correlated with a defect in the so-called post-replication end-capping function of Cdc13 (Fig 4; Table 1). Third, the initial telomere-linked unstable chromosomes progress to form the different types of rearrangements we have previously observed, plus a novel one at an internal site with synthetic TG telomere repeats (Figs 5 and 6). We discuss key features of our findings, followed by a speculative model.

## The nature of Cdc13 function and a replication defect

Clearly understanding the intricacies of Cdc13 function is key to understanding instability. Cdc13 acts in a heterotrimer with Ten1 and Stn1. Because a *cdc13^F684S stn1^T223,S250A* double mutant is more unstable than either single mutant, it is likely the heterotrimer that needs to function to suppress instability. Cdc13's interaction with telomerase is unlikely to be involved, as a telomerase defect takes many generations to cause instability [47,54], whereas a defect in Cdc13 initiates instability in a single generation (this study).

A DNA replication defect in *cdc13* mutants is likely important, as we find that inactivation of Cdc13 in S phase causes instability, but not inactivation in G2/M cells (Fig 4). That inactivation of Cdc13 in G2/M cells does not generate ssDNA nor cause instability calls into question the extent to which Cdc13/CST is really capping the chromosome end independent of replication. We suggest that during replication, CST could assist lagging-strand synthesis by acting at either the end of the chromosome or at the replication fork. We favor the hypothesis that Cdc13 acts from the end of the chromosome rather than from the replisome since we do not detect chromosome instability originating from the interstitial 126 bp TG repeat. It should be noted that a replication error in the interstitial TG repeat would be rescued by the oncoming fork, an unavailable option at the telomere. We therefore cannot rule out the possibility that Cdc13 acts at internal loci, only that *cdc13*-dependent unstable chromosomes are not generated there.

We do find some interaction between HU treatment and *cdc13* instability, implying a defect in DNA replication causes instability; but how HU treatment affects CST function at the telomere is unclear. HU treatment is known to prevent the firing of late origins [75], so perhaps the replication forks are more fragile at the telomere, which are the last region to be replicated in *S. cerevisiae*. In this context, further stress from the *cdc13* deficiency may result in increased chromosome instability.

## What happens to a damaged chromosome when Cdc13 is restored?

The structure and fate of chromosomes that incur damage due to a Cdc13 defect, then remain damaged (to form unstable chromosomes) when Cdc13 is resynthesized is a question central to understanding unstable chromosomes. There are few studies of how Cdc13-defective and arrested cells recover [76]. How might a *cdc13*-induced "end-gap" be repaired? We note that mutants in post-replication repair (PRR) (e.g. *rad18Δ*) do not synergize with a *cdc13* defect, suggesting that whatever ssDNA gap structures are present are not repairable by PRR. Whether an unstable chromosome is formed before or after Cdc13 resynthesis is also unclear. We posit that extensive ssDNA may engage strand-invasion reactions, not permitting completion of failed lagging strand synthesis.

## Before the model: Issues regarding CAN1 and Exo1-dependent degradation

There are several issues concerning the position of *CAN1*, degradation by Exo1, and unstable chromosomes that affect models of instability. Defects in Cdc13 cause extensive degradation that is Exo1 dependent; how far does the degradation proceed, and is the *CAN1* gene rendered single stranded? Given that Exo1 degrades dsDNA to ssDNA at about 4 kb/hour [61], and that *CAN1* resides 25 kb from the telomere, it is unclear to what extent the *CAN1* gene becomes single-stranded in 4 hours in that first cell cycle. Second, we posit that in one cell cycle, only one of the two sisters might lose integrity of the *CAN1* gene. It seems reasonable to postulate that one sister chromosome (product of leading strand synthesis) is fully intact when Cdc13 is defective; though this is speculative. If so, the cell with *cdc13*-induced damage should be Can$^S$ with one intact *CAN1* gene. Apparently, there is sufficient phenotypic lag to enable generation of a Can$^R$ and a Can$^S$ cell from one cell division in media with canavanine. And finally, what occurs in *cdc13$^{F684S}$ exo1Δ* mutant cells to possibly convert an unstable chromosome to a recombinant (Table 1)? The more limited ssDNA gaps in *exo1* mutants seem to favor recombination. The position of *CAN1* relative to the telomere, rate of Exo1-dependent degradation, the fate of sister chromosomes, and the role of smaller versus larger gaps are issues remaining to be resolved to permit a clearer model.

## A model of Cdc13-induced instability and an unstable chromosome

Despite the uncertainties of *cdc13*-induced instability, we know enough to propose the specific model in Fig 7. In this model, a defect in CST *post* replication has no consequences for chromosome stability. A defect in CST *during* S phase, however, generates a G2 cell with one intact and one defective sister chromosome from leading and lagging strands, respectively [30,77]. The damaged sister attempts repair by strand invasion and replication. If repair occurs using the intact sister (not drawn), no genetic change arises. If repair occurs using the homolog as the template, the outcomes depend on what sequences pair during strand invasion. Invasion telomere-proximal to *CAN1* would result in no genetic change (not drawn). Invasion centromere-proximal to *CAN1* would result in loss of *CAN1* and a Can[R] cell. Invasion centromere-proximal to *CAN1* might benefit from ssDNA incurring an internal "nick". After invasion, either replication proceeds to the telomere to form a recombinant, or replication aborts, the invaded strand seeks in vain a second end capture. With no second end capture possible, the unstable chromosome undergoes progressive degradation, repeated attempts of strand-invasion and dissociation, forming other forms of chromosome products.

This model is essentially the Break Induced Replication (BIR) pathway, attractive given a link between telomeres and BIR [78–80]. Determining whether failed BIR is indeed responsible for creating the initial unstable chromosome and its progression is a future research goal. Cycles of instability in sectored colonies bear some similarity to "cascades of instability" reported for a telomere-linked unrepairable DSB in a Chr III disome [81], a system used to decipher molecular events of BIR. This model poses a plethora of predictions yet to be tested, including the possible role of adaptation in cell cycles resuming with a linear unstable chromosome and the role of the distance of *CAN1* from the telomere, etc.

## Progression of telomeric unstable chromosomes

We suggest that an unstable chromosome forms near the telomere and from there progresses to assume one of many fates; recombination in various regions along the chromosome, chromosome truncation at an internal TG sequence, dicentric formation via fusion of inverted repeats at the 403 site, or chromosome loss. We suggest these forms may arise through progressive degradation of a linear chromosome by Exo1 [55]. Given the "Failed BIR Model" in Fig 7, we suggest that at any point during degradation, strand invasion into the homolog may arise, resulting in a Class I recombinant if invasion and successful replication occurs early, or Class II if it occurs late. When degradation proceeds to the $^{281}TG_{126}$ site, a truncated chromosome forms (the cell is still viable because there are two copies of Chr VII). When degradation subsequently proceeds to the 403 site, the inverted repeat sequences are exposed as ssDNA and an intra-sister fusion occurs to form a dicentric. Exactly how telomeric ssDNA is converted to recombinants, whether it undergoes a nick as shown in the Fig 7, and whether BIR is involved are yet to be resolved.

## Materials and methods

### Yeast strains

All yeast strains used in this study were derived from the A364a TY200 disome strain previously described (S2 Table; [44–46]). The TY200 wild type Chr VII disome strain is *MATα +/hxk2::CAN1 lys5/+ cyh^r/CYH^S trp5/+ leu1/+ cenVII ade6/+ +/ade3, ura3-52*. The endogenous *CAN1* gene on Chr V was mutated and inserted on one copy of Chr VII in place of *HXK2*. Additional strains were generated by LiAC/ssDNA/PEG transformation of TY200 with

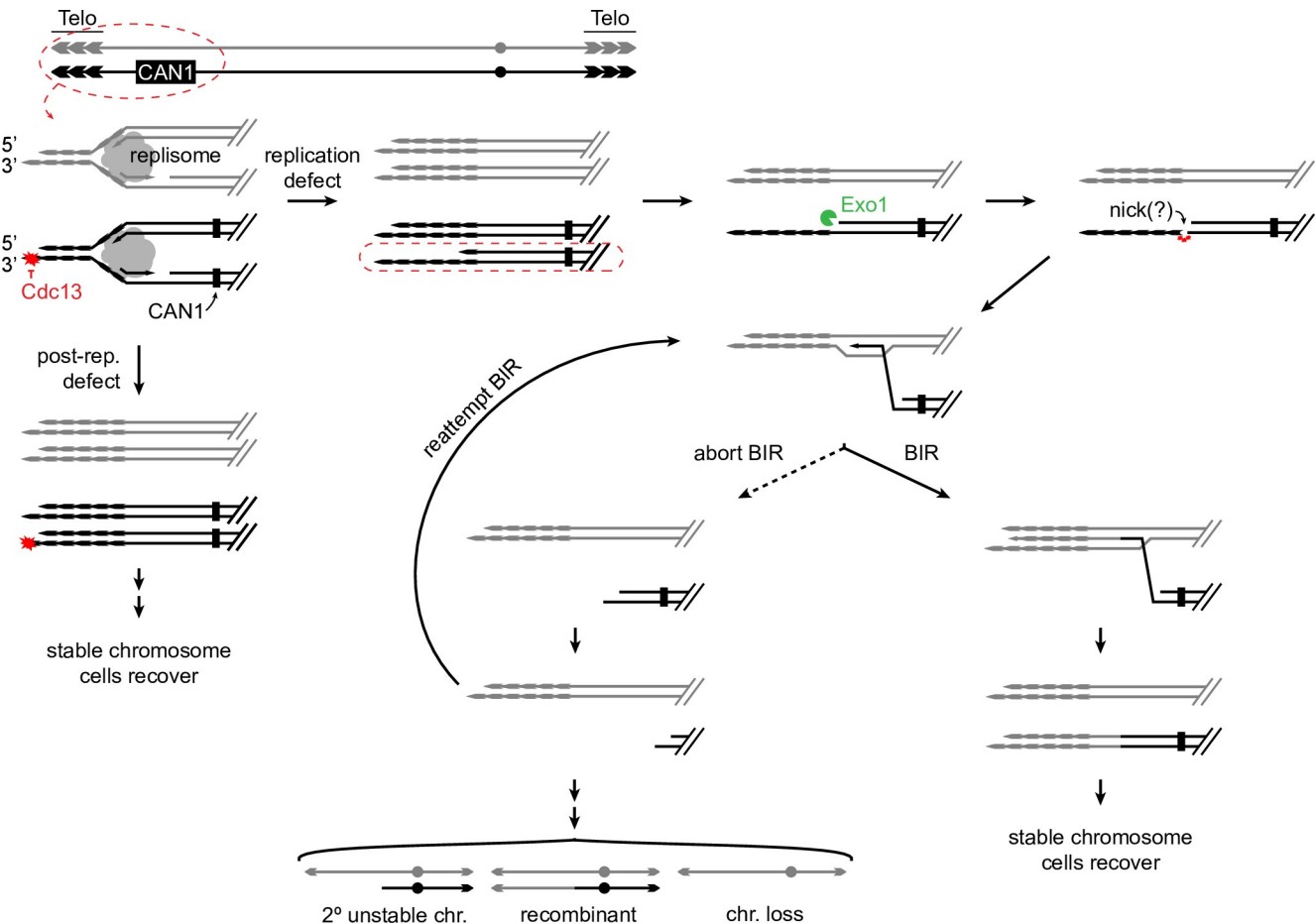

**Fig 7. Model for formation of unstable chromosomes in the telomere.** A Cdc13 defect leads to a defect in replication of one homolog. The defect leads to extensive ssDNA of the lagging strand template sister, and a complete leading strand template sister. The lagging strand template is nicked and acts as the parental primer strand, which invades an intact homolog in single copy sequence. Two outcomes are possible: replication to the end of the chromosome or re-initiation of BIR. BIR may again abort, in which case the parental primer 3' may shorten, invade the homolog, and attempt replication again.

DNA fragments or plasmids. Strains were verified by genetic analysis and/or PCR. There were at least two independent strains made and analyzed for each mutant reported.

The various *cdc13* mutant strains was made by either Margherita Paschini or in house with a plasmid carrying the corresponding *cdc13* mutation via a "pop-in, pop-out" mechanism (pVL5439 (*cdc13^{F684S}*); pVL7013 (*cdc13^{Y556A,Y558A}*); gifts from V. Lundblad; [49]). Double mutants with *cdc13^{F684S}* were constructed by transforming PCR amplifications of *KanMX4*-marked genes replacements from the Gene Deletion Library into *cdc13^{F684S}* strains using primers that flanked the coding region. It should be noted that Chr VIII and V have recombined in one isolate of *cdc13^{F684S}*. We do not believe that these alterations affect the frequency of chromosome instability since neither copy of Chr VII was affected and both variants of *cdc13^{F684S}*, with and without the altered karyotype, had the same frequency of all three types of chromosome instability. Double mutants integrated into the *cdc13^{F684S}* isolate with the altered karyotype are marked in the table in S2 Table.

The *URA3* proxy strains were generated as previously described [46]. Briefly, the *URA3* gene fragments were linked to genes encoding drug resistance in two cassettes: RA:*NAT1*:RU

and A3:*KanMX4*. These were inserted into plasmids containing ~500 bp of sequence homology to flanking sites of the 403 kb (RA:*NAT1*:RU) or 535 kb (A3:*KanMX4*) regions of Chr VII. Plasmids were digested with the appropriate restriction enzymes to free the targeting fragment and *cdc13*$^{F684S}$ cells were transformed and selected for drug resistance. Correct insertions were verified by PCR and genetic analysis. In genetic analyses we verified that insertions were linked to *CAN1* (the changes were in the bottom homolog in Fig 1).

The $^{281}$TG:*URA3* strains were generated by PCR amplification of a 126 bp TG track from telomere IL from a pRS406:TG$^{126}$ plasmid. The pRS406:TG$^{126}$ plasmid was created by subcloning the TG fragment from a PCR 2.1 vector (gift from V. Lundblad) into pRS406 with restriction enzymes. The TG$^{126}$ fragment was inserted into the genome by amplifying the construct with ends homologous to the target region (45 bp of homology). Cells were then transformed and selected for Ura$^+$. Isolates were verified by PCR with primers outside of the targeted region. Further genetic analysis was performed to verify that insertions were integrated into the *CAN1* homologue of Chr VII.

The $^{122}$*KanMX4* marker was integrated into *cdc13*$^{F684S}$ by transforming *cdc13*$^{F684S}$ with DNA fragments containing the *KanMX* gene flanked by 45 bp of homology to the Chr VII 122 kb locus. DNA fragments were synthesized by PCR amplification of the *KanMX4* gene from the pRS400 plasmid with primers containing 45 bp of homology to the target sequence. The *KanMX4* gene replaced Chr VII 121500 bp– 122100 bp. Correct insertion of the *KanMX4* construct was confirmed via PCR with primers outside of the targeted region, and the integration of the construct into the *CAN1* homologue of Chr VII was confirmed by genetic analysis (Can$^R$ colonies became sensitive to G418/geneticin).

The following drug concentrations were used whenever drug selection was appropriate: canavanine (Can; 60 μg/mL), G418/geneticin (100 μg/mL), hygromycin B (Hyg; 300 μg/mL), and nourseothricin (Nat; 50 μg/mL).

## Chromosome instability assays

Genetic analysis determined the frequency of sectoring, round recombinants, and chromosome loss by a slightly modified version of the previously described instability assay [44]. Briefly, strains were struck to minimal media (Min + uracil) at 25˚C to select for retention of both Chr VII homologs. Single cells were then plated to rich media (YPD, 2% dextrose) and grown for 2–3 days at 30˚C to form colonies and allow instability to arise. Once colonies were formed, individual colonies were suspended in water, cell concentration was determined by counting cells with a haemocytometer, and a known number of cells were plated to selective media and grown at 25˚C to measure chromosome instability.

To determine the frequency of chromosome loss cells were plated to selective media containing canavanine (60 μg/mL) and all essential amino acids except arginine and serine. Cells were grown at 25˚C for 5–6 days to allow for colony formation. Chromosome loss was determined by replica-plating to identify Ade$^-$ colonies (Can$^S$ Ade$^-$). The frequency of chromosome loss was calculated by determining the ratio of the number of Can$^R$ Ade$^-$ colonies by the total number of cells originally plated.

To determine the frequency of unstable chromosomes and recombination, cells were plated to canavanine plates that also lacked adenine (in addition to arginine and serine). Cells were grown for 5–6 days at 25˚C and then colonies were scored based on morphology (sectored or round, Fig 1B). The frequency of sectored (unstable chromosomes) and round (recombinants) was calculated by determining the ratio of the number of Can$^R$ Ade$^+$ sectored or round colonies to the total number of cells originally plated. Medians and interquartile ranges were calculated from analysis of at least 6 colonies grown on rich media that were then plated to selective

media. Unless otherwise stated, statistical tests were performed using the Mann-Whitney U method; individual instability frequencies and exact $P$ values are reported in S3 Table.

## Phenotypic analysis of altered chromosomes

*Single Colony Lineage Analysis*: Cells from a single round or sectored Can$^R$ Ade$^+$ colony was isolated and suspended in water. Approximately 200 to 300 cells were then plated to rich media (YPD) and grown for 2–3 days at 25°C to form colonies, each from a single cell. The phenotypes of each colony were determined by replica-plating to synthetic media, each lacking one essential amino acid, or media containing a drug (cycloheximide, 5 μg/ml). Cells were grown for 2–3 days at 25°C, and growth on each selective plate was assessed. The proportion of Class I or II recombinants per region was calculated by dividing the number of genetic recombinants with the total number of recombinant colonies assayed. Colonies that lacked all the markers were scored separately as chromosome loss.

*Pooled Colony Lineage Analysis*: Cells from approximately 50 round or sectored Can$^R$ Ade$^+$ colonies were pooled and suspended in water. The retention and loss of individual genetic markers were analyzed as done in the Single Colony Lineage Analysis.

## Single cell cycle assay

Cells were grown to mid-log overnight at 25°C in 100 ml of YPD. Cells were counted via hemocytometer to confirm that they were in mid-log ($2-6 \times 10^6$ cells/ml). Once the cells were at the appropriate concentration, hydroxyurea was added to 20 ml of culture (0.2 M HU) and the culture was incubated at 25°C for 3 hours. Cells were then washed twice with H$_2$O and resuspended in 20 ml of YPD ($t_0$). The culture was then split; half of the culture was incubated at 37°C while the other half remained at 25°C. The cultures were incubated at either 37°C or 25°C for a total of 4 hours. To measure chromosome instability, 1 ml of culture was taken from both cultures every hour after the temperature shift, as well as from the $t_0$ time point. The 1 mL aliquots were sonicated, counted, and plated to CAN and CAN-ade for the instability assay within 30 minutes of collection (instability assay described above). Cells were also plated to selective plates lacking arginine to measure cell viability. Viability was measured after 1 day of growth by counting the number of viable microcolonies versus single cells. For the control, cells were never exposed to HU and were instead shifted to either 37°C or 25°C immediately after they were confirmed to be in mid-log.

Samples for FACS were taken at mid-log, after HU was washed out (if applicable), and then at 0:15, 0:30, 0:45, 1:00, 2:00, 3:00, and 4:00 hours after cells were shifted to either 37°C or 25°C. The section below describes the FACS protocol.

To inactivate *cdc13$^{F684S}$* during S phase, cells were grown to mid-log overnight in 100 ml of YPD at 25°C. Once in mid-log growth, a 1 ml aliquot was taken for the instability assay and 10 ml of culture was incubated at 37°C. After 4 hours at 37°C another 1 ml aliquot of cells was taken for the instability assay.

To limit *cdc13$^{F684S}$* inactivation to G2, cells were grown to mid-log overnight in 100 ml of YPD at 25°C. Once in mid-log, nocodazole was added to 20 ml of culture (10 μg/ml, 1% DMSO) and the culture was incubated at 25°C for 3 hours. After incubation, cells were washed twice with water and resuspended in 20 mL of fresh YPD ($t_0$). The culture was then incubated at 37°C for 4 hours. Aliquots of cells were taken at $t_0$ and after 4 hours at 37°C for the instability assay. To hold cells in G2/M after the temperature shift additional nocodazole was added (10 μg/ml; 1% DMSO).

As done previously, the 1 mL aliquots for the instability assay were sonicated, counted, and plated to CAN and CAN-ade for the instable assay within 30 minutes of collection. Cells were also plated to selective plates lacking arginine to measure cell viability.

Samples of FACS analysis were taken at mid-log, after nocodazole was washed out (if applicable), and at 0:30, 1:00, 2:00, 3:00, and 4:00 after the cells were shifted to 37˚C. The FACS protocol is described in the following section.

### FACS analysis

Cells were pelleted from 1 mL of log YPD culture and fixed in 70% ethanol overnight at 4˚C. Ethanol was removed and cells were resuspended in 50 μl of 50 mM sodium citrate (pH 7.4) and sonicated at low power (8 s at 20% power). Cells were pelleted again and resuspended in 1.0 ml of 50 mM sodium citrate with 0.25 mg/ml of RNaseA for 1 hour at 50˚C. 25 μl of proteinase K solution (20 mg/ml proteinase K in 10 mM Tris pH 8.0, 1 mM $CaCl_2$, 50% glycerol) was subsequently added and the cells were incubated at 50˚C for an additional hour. Cells were then pelleted and resuspended in 1.0 ml of 50 mM sodium citrate containing 2 μM Sytox green (Invitrogen). Cells were left to incorporate the dye overnight at 4˚C. Samples were scanned using a Attune Acoustic Focusing Flow Cytometer (Invitrogen).

### In-gel hybridization and Southern analysis

750 ng– 1 μg of XhoI digested genomic DNA was subjected to agarose gel electrophoresis (0.7% agarose) for 16 hours at 60 V. The gel was placed on a double-layer of Whatman paper with a piece of plastic wrap over top and mounted on a Bio-rad (model 583) gel drier. Drying was carried out for ~20 minutes at room temperature. The dried gel was sealed in a plastic bag and hybridized with at least 1 million CPMs of γ-ATP32 end-labeled oligonucleotide 5'-CCCA CCCACCCACCACACACACCCACACCC-3' in in-gel hybridization buffer overnight in a 37˚C water bath. After removal of excess hybridization buffer, the gel was washed 3–4 times for 30 minutes in 0.25X SSC at room temperature with agitation, sealed in a bag, and exposed to Amersham Hyperfilm MS or phosphor cassette. After, the in-gel was subjected to a denaturing Southern blot in order to visualize the total telomeric DNA. The Southern blot was performed as follows: 10 minutes in 0.25M HCl, 45 minutes in Southern denaturing solution (1.5M NaCl, 0.5M NaOH), 5 minutes 0.4M NaOH. It was then transferred to a Hybond-XL nylon membrane. After transfer, the membrane was prehybridized for 1 hour in Church buffer and then hybridized overnight at 65˚C with a radiolabeled probe that hybridizes to telomeric repeats. The membrane was washed for 20 minutes at room temperature in 2X SSC and exposed on a phosphor cassette.

### Pulsefield gel electrophoresis and Southern analysis

To identify altered chromosomes, pulsefield gel electrophoresis and Southern blotting for chromosome VII were performed as described [44]. Southern hybridization was performed using a $P^{32}$-labeled probe to Chr VII sequences 503875 bp– 505092 bp as previously described [47].

## Supporting information

**S1 Fig. Diagram of the instability assay.** (A) Cells are plated to rich media for 2 days at 30˚C (semi-restrictive temperature for *cdc13^F684S^*), then individual colonies are plated to canavanine-containing media for 5 days at 25˚C to select for chromosomal rearrangements. Instability is scored by comparing the number of "sectored" or "round" colonies vs the number of

cells originally plated.

(B) Asynchronous cells are grown at 25˚C then shifted to 37˚C for 1-4h. Then ~$10^5$ cells are plated to canavanine-containing media for 5 days at 25˚C to select for chromosomal rearrangements. Instability is scored by comparing the number of "sectored" or "round" colonies vs the number of cells originally plated.

(TIF)

**S2 Fig. Sectored colonies contain multiple genotypes.** (A) Histogram for the number of genotypes in sectored (red) and round (orange) colonies, n = 20 (n = 10 colonies each for sectored and round; experiment done at 30˚C).

(TIF)

**S3 Fig. Inserting the TG$_{1-3}$ repeat into the 484 kb locus does not change the distribution of Class I recombinants in round colonies, but does change in distribution in sectored colonies at a semi-restrictive temperature (30˚C).** (A) Diagram for the TG repeat (green box) is insertion in the 484 kb locus.

(B) Frequency of round (*top*) and sectored colonies (*bottom*) from an unmodified $cdc13^{F684S}$ and $cdc13^{F684S\ 484}TG_{1-3}$ are not altered. The median is shown (n > 12; * < 0.01; Mann-Whitney U).

(C) Distribution of genotypes from round (Class I recombinants) and sectored colonies (Class II recombinants) from $cdc13^{F684S}$ with no insert or with the TG$_{1-3}$ insert. The average percentage and standard deviation for 3 independent experiments are shown.

(TIF)

**S4 Fig. Inserting the TG$_{1-3}$ repeat into the 281 kb locus does not alter the frequency of unstable chromosomes, recombination, or chromosome loss.** (A) Frequency of instability for sectored, round, and chr. loss colonies from an unmodified $cdc13^{F684S}$, $cdc13^{F684S}$ *URA3*, and $cdc13^{F684S\ 281}TG_{1-3}$. The median is shown (n > 12; * < 0.01; Mann-Whitney U; experiment done at 30˚C).

(TIF)

**S5 Fig. Expanded model of the unstable chromosome's fate on a cellular level.** (A) Complete schematic of how diversity arises from one unstable chromosome. An unstable chromosome replicates, then various descendent chromosomes rearrange in subsequent cell divisions. Further explanation in text.

(TIF)

**S6 Fig. $cdc13^{F684S}$ activity in S and inactivity for either only G2 or only G2-T does not result in unstable chromosomes.** (A) Schematic for arresting cells in either G2/M or T phase. Cells were grown at 25˚C until they reached mid-log, then were incubated with nocodazole (noc) for 3 hours. Cells were then washed (0h) and the culture was split. Both halves were incubated at 37˚C for 4 hours, one with (4h + noc; *top*) and without (4h –noc; *bottom*). At asy, 4h + noc, and 4h –noc cells were collected for DAPI staining and the instability assay.

(B) Nuclear morphology based on DAPI staining of chromosomes. The proportion of $cdc13^{F684S}$ *cdc15-2* cells with each morphology for the different arresting protocols is shown. Blue: cells grown at 25˚C; red: cells grown at 37˚C. Mean and standard deviation from at least 6 independent experiments are shown.

(C) Fold change in the frequency of sectored, round, and chr. loss colonies (4h/0h) from *cdc15-2* CDC13$^+$ and $cdc13^{F684S}$ *cdc15-2* with (n = 6; 6) and without (n = 6; 8) additional nocodazole added after the initial noc arrest. The fold change between 4h and 0h remained low

despite the arresting conditions ($^* < 0.05$, Mann-Whitney U).
(TIF)

**S7 Fig. ssDNA generated in $cdc13^{F684S}$ during G2/M probably comes from escaped cells in the next cell cycle.** (A) Uncropped image of the non-denaturing in-gel hybridization in Fig 4E. Note that ssDNA is generated in $cdc13^{F684S}$ after nocodazole arrest at 25˚C and release into 37˚C (lanes 13–18; not depicted in Fig 4E).
(B) Additional non-denaturing in-gel hybridization using a CA oligonucleotide probe with XhoI-digested DNA from the indicated samples (*top*). To control DNA input, the gel was denatured and transferred to a nitrocellulose membrane and hybridized with a telomeric probe (*bottom*). Cells were grown in the indicated conditions for 3 hours.
(TIF)

**S8 Fig. $cdc13^{F684S}$ cells proceed through 1–3 cell divisions before dying on canavanine.** Density curve for the colony size of $cdc13^{F684S}$ ($t_0$ cells from Fig 3A *top* grown on plates with (*top*) or without canavanine (*bottom*). Colony size was scored at 24h (purple) and 48h (blue). Dashed red line: the average number of buds per colony for cells grown with canavanine (mean = 4.01 buds). Experiment done at the permissive temperature (25˚C).
(TIF)

**S9 Fig. Unstable chromosomes can resolve by forming truncations at an internal TG repeat at the semi-permissive temperature (30˚C).** (A) The $^{281}$URA3 or $^{281}$TG repeat (green box) inserted between *LYS5* and *CYH2* as in Fig 2B. Red line indicates the probe binding site.
(B) Distribution of Class II recombinants from sectored colonies from $cdc13^{F684S}$ with *URA3* or *URA3*::TG$_{1-3}$ inserted into the 281 kb locus. The average percentage and standard deviation for 3 independent experiments are shown.
(C) Proportion of Lys$^-$ Cyh$^S$ cells that have additionally lost *URA3* from sectored and round colonies. *P* values were calculated with a *t* test.
(D) Southern blot of pulse-field gels from Lys$^-$ Ura$^-$ Cyh$^S$ cells from sectored or round colonies. A Chr VII centromere-linked probe was used to label chromosome VII. The upper band corresponds to the normal Chr VII size (1090 kb) while the lower band corresponds to a truncation at 281 kb (810 kb; localizes with Chr II, 813 kb).
(TIF)

**S10 Fig. Inserting the TG$_{1-3}$ repeat in the reversed orientation does not result in either an increase in instability or in a "spike" in TG-associated Class I or Class II recombinants at a semi-permissive temperature (30˚C).** (A) The TG repeat (green box) in the 281 kb in the reversed orientation (TG-rich strand acting as the template for leading strand replication).
(B) Frequency of round (*top*) and sectored colonies (*bottom*) from an unmodified $cdc13^{F684S}$ (white) and $cdc13^{F684S}$ $^{281}$TG$_{1-3}$-rev (green) are not altered (n > 12; $^* < 0.01$; Mann-Whitney U).
(C) Distribution of genotypes from round (Class I recombinants) and sectored colonies (Class II recombinants) from $cdc13^{F684S}$ with no insert, with $^{281}$TG$_{1-3}$, or with $^{281}$TG$_{1-3}$-rev. The average percentage and standard deviation for 3 independent experiments are shown ($P = 0.01$; one sample *t* test).
(TIF)

**S1 Table. Median frequencies of chromosome instability in additional $cdc13^{F684S}$ mutants at 30˚C.** Median values and IQR, in []s, are reported.
In bold, statistically significant ($P < 0.01$; Mann-Whitney U) fold change between single mutants and Wild Type (CDC13$^+$) or between $cdc13^{F684S}$ *mutX* and $cdc13^{F684S}$

Normalized median frequencies are reported in ()s (normalized to CDC13$^+$).
(DOCX)

**S2 Table. Saccharomyces cerevisiae strains used in this study.** [a] All strains are disomic for Chr VII and are derivatives of TY200 MATα +/hxk2::CAN1 lys5/+ cyh$^r$/CYH$^S$ trp5/+ leu1/+ Centromere ade6/+ +/ade3, ura3-2.
[b] The URA3 module, TG$_{1-3}$ repeat, and/or selective markers were integrated into the *CAN1* homologue of Chr VII. The insert locations are notated (kb from the left telomere of Chr VII).
* Double mutants integrated into *cdc13$^{F684S}$* with altered chromosomes V and VIII.
(DOCX)

**S3 Table. Exact measurements and *P* values for this study.** Exact chromosome instability measurements and *P* values (Mann-Whitney U).
(XLSX)

# Acknowledgments

We thank Vicki Lundblad and Margherita Paschini for the *cdc13$^{F684S}$* and *cdc13$^{Y556A,Y558A}$* mutants in addition to helpful discussions, plasmids, and construction of strains. We thank Shang Li for plasmids. We also thank Lisa Shanks, Tracey Beyer, and Peter Vinton for frequent discussions of this work.

# Author Contributions

**Conceptualization:** Rachel E. Langston, Ted Weinert.

**Funding acquisition:** Rachel E. Langston, Raymund J. Wellinger, Ted Weinert.

**Investigation:** Rachel E. Langston, Dominic Palazzola, Erin Bonnell.

**Methodology:** Rachel E. Langston, Erin Bonnell, Ted Weinert.

**Project administration:** Ted Weinert.

**Supervision:** Raymund J. Wellinger, Ted Weinert.

**Validation:** Rachel E. Langston.

**Visualization:** Rachel E. Langston.

**Writing – original draft:** Rachel E. Langston, Ted Weinert.

**Writing – review & editing:** Rachel E. Langston, Erin Bonnell, Raymund J. Wellinger, Ted Weinert.

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
