## [Decision Letter · Decision Letter 0]

28 Oct 2019

Dear Dr Weinert,

Thank you very much for submitting your Research Article entitled 'Loss of Cdc13 causes genome instability by a deficiency in replication-dependent telomere capping' to PLOS Genetics. Your manuscript was fully evaluated at the editorial level and by independent peer reviewers. The reviewers appreciated the attention to an important problem, but raised some concerns about the current manuscript. Most of the reviewers concerns relate to the presentation and interpretation of the data and would require only revisions to the text. In addition, I do recommend testing the cdc13 sae2 rad9 mutant to distinguish between resection and hairpin opening functions of Sae2, as suggested by Reviewer 1. Based on the reviews, we will not be able to accept this version of the manuscript, but we would be willing to review again a much-revised version. We cannot, of course, promise publication at that time.

If you decide to revise the manuscript for further consideration at PLOS Genetics, please aim to resubmit within the next 60 days, unless it will take extra time to address the concerns of the reviewers, in which case we would appreciate an expected resubmission date by email to plosgenetics@plos.org.

[LINK]

We are sorry that we cannot be more positive about your manuscript at this stage. Please do not hesitate to contact us if you have any concerns or questions.

Yours sincerely,

Lorraine S. Symington

Associate Editor

PLOS Genetics

Gregory P. Copenhaver

Editor-in-Chief

PLOS Genetics

Reviewer's Responses to Questions

**Comments to the Authors:**

Reviewer #1: The authors investigated the role of Cdc13 in maintaining genome stability. By using the temperature sensitive cdc13-F684S, they show that the passage through S phase is necessary to generate chromosome instability (recombinants, chromosome truncations, dicentrics, and/or loss) in cdc13-F684S mutant. Generation of unstable chromosomes requires Exo1, but not homologous recombination and non-homologous end joining. The authors propose a model where Cdc13 dysfunction impairs semi-conservative replication at the telomere that leads to ssDNA and unstable chromosomes.

The manuscript is interesting and the experiments are well done. The major finding that instability in cdc13 mutants is replication dependent is novel and important.

I have some suggestions that might help to improve the understanding of the manuscript:

1. Lines 157-159: the interpretation of the synergistic increase in cdc13 stn1 double mutant is difficult to understand. As Cdc13 and Stn1 interact, it could simply be that each single mutation does not fully impair Cdc13 and Stn1 function but, when they are both present in the same cells, this lead to a more severe defect of CST function that could explain the synergistic effect.

2. Lines 240-248: Since telomerase dependency was not tested, I was asking whether it is necessary to distinguish lagging strand synthesis after telomerase or independent of telomerase…

3. Pag 252: delete the sentence “A method…” or explain why it is not possible to inactivate Cdc13 only in S phase (what about releasing cells from a G1 arrest at 37°C?).

4. Pag 329-345: It is quite clear that the lack of Sae2 suppresses instability in cdc13 mutant, but the effect on rfa1-t33 is difficult to understand. The lack of Sae2 impairs hairpin cleavage but it also increases Rad9 binding at DSBs. Since the lack of Rad9 increases instability in cdc13 mutants, it could also be that the lack of Sae2 suppresses instability by increasing Rad9 binding. This could be tested in cdc13 rad9 sae2 triple mutant to see whether suppression depends on Rad9.

In any case, the authors propose that “it is unlikely that the hairpin mechanism is responsible for chromosome instability”. However, the model includes hairpin cleavage by Sae2. Furthermore, at line 424 they “suggest a role for cleavage of hairpins…”. This is confusing…

Reviewer #2: Langston et al investigate the timing of action of a cdc13ts allele and the types of chromosome instability that are found as a consequence of loss of cdc13 function under a variety of different conditions that depend on the presence of active Cdc13 is S and G2/M phases or only in the G2/M phase. The authors find that the S phase activity is required for the cdc13ts defects and form two class of recombinants as well as chromosome loss events. The authors demonstrate through dependence on Exo1 and through identification of single-stranded telomeric DNA in G2/M that single-stranded regions are critical for the formation of recombinants. The recombinants are favored at telomere proximal regions and are not simply due to the presence of TG repeats that, when present at distal sites, do not change the products that are formed with the exception of a truncation at 482 kb.

These data are highly important as they demonstrate the need for DNA replication for Cdc13 activity that prevents recombination and chromosome loss. The work is thorough and, for the most part, experiments are well conceived. The quality of the data is, in general, very high. The paper could have been written more clearly and there are grammatical mistakes throughout. More importantly, there are explanations and interpretations that are oversimplified or omitted that require clarification and stipulation. The specific major and minor criticisms are enumerated below.

Major criticisms:

1. There are several stipulations regarding the compared shifts into various media.

These are based on several technical limitations. First, cells that shift the cdc13 ts after growth to nocodazole (Figure 4) may have already passed through part of the G2 phase. The possibility for some activity in the G2 phase still remains. This finding needs to be clearly stipulated. Furthermore, the authors do not discuss whether the cdc13ts arrests before or after the nocodazole point. In fact, it is impossible to ascertain since after the shift to 37°C nocodazole is again added. The fact that the phenotype depends on growth at 37°C, however, argues for an effect of the mutant so it probably acts at or near the nocodazole point.

2. T The comparison between the shifting experiments are not always appropriate. In Figure 3, a shift of asynchronous cells to 37, allowing both S and G2/M arrest, is compared to cells synchronized and maintained with HU, a DNA synthesis inhibitor, followed by shift to 37C. The former yields less of an increase in instability than the latter, leading to the conclusion that incubation with HU before S phase participates the subsequent S phase activity of Cdc13. What is important is that both of these schemes that lead to S phase involvement. A better comparison would be arrest with alpha factor before wash-out before shift-up in growth that would also be presumably S phase to continue until G2/M arrest. Arrest of cells in HU could simply increase the frequency of recombinants by limiting the cells analyzed to arrested cells rather than to cells at any cell cycle stage. This lack of parallel approach does not take away from the interest in HU, although it is not central to the paper. The experimental protocol is simply not parallel. The comparison is also not appropriate in Figure 4 were asynchronous cells are compared to nocodazole arrested cells before shift-up. Again, the comparison between asynchronous cells and synchronized cells are not parallel. Lines 236-238 are overstated.

3. The characteristics of the ts allele is that it is degraded very rapidly at the restrictive temperature. This indicates that any events after shifting down to 25°C does not involve immediate renaturation, but rather re-synthesis of the protein. This should be noted, although it does not affect the interpretation of the data, it may influence the timing of recovery. Hence on line213, “We then reactivated the protein” should probably be replaced by “We then resynthesized the protein”.

4. The authors should draw or explain Class I and Class II recombinants after they are first mentioned.

5. How are the plots of the expectation of random recombination determined?

6. Line 159: How do the authors explain the lack of stn1 mutant phenotype in chromosome instability?

7. Lines 239-248. In the logical argument for two among the four possible mechanisms of Cdc13 activity, telomerase activity is eliminated and, logically, so is lagging strand activity associated after telomerase extension. However, please consider the possibility (textually) of lagging strand synthesis facilitation in the absence of telomerase extension.

8. Line 531-533: Please describe the alterations. The vague description is not acceptable for the acceptance of the corresponding data.

9. Tables 1 and 2. The meaning of the bold indication of significance is incomplete. Why are chromosome loss events not listed as significant?

10. Why do statistical tests vary between t tests and Rank Sum in different experiments? Please explain.

11. Figure Legends have multiple errors and omissions that confuse the reading of the Figures:

a. Line 899: replace with “heterozygous markers”.

b. Line 901: What do” replicas” refer to?

c. Asterisks are defined in Figure 1 Legend but are not present in Figure 1.

d. The temperature of growth in each experiment must be noted in the Figure Legends for interpretability.

e. The cdc15-2 mutation must be introduced in the text associated with Figure 3, rather than Figure 4, since it is used in the former figure.

Minor criticisms: Textual recommendations

1. Line 66, Please define Cdc13/CST.

2. Line 70, change “overhand” to “overhang”.

3. Line 91, please give background on Sae2 function.

4. Line 106, change “live” to “grow on canavanine-containing media”.

5. Lines 112-114, the description is clumsy, please rewrite.

6. Throughout the manuscript, the authors compare or list phenotypes and the type of aberrant events in sentences creating confusion. For example, round Can R Ade+ with recombinant forms and Can R Ade- with chromosome loss events are intermixed Choose either the event or the phenotype in any given sentence.

7. Change all ‘loss’ events to ‘chromosome loss’ events.

8. Example: line 119: change to “unchanged cell dies on canavanine due loss of the dominant Can1S gene, form unstable…...and chromosome loss in CanRAde2- colonies. Another example in line 141 to 142.

9. Delete “Two additional features of this model that”. It is redundant. “The unstable chromosomes give rise to both recombinant and chromosome loss. There are two class of recombinants. We theorize….

10. Throughout capitalize CDC13 when mentioned as a gene.

11. Line 131, Change “introduced “to “integrated”

12. Line 139: change to “and plated onto selective media as an assay for chromosome instability.”

13. Line 149: In the absence of evidence, change “many, if not all” to “multiple”.

14. Line 163: There is a general enthusiasm in the use of colons and “ yet it could” phrases in the text. In almost every case, a break between two sentences would be more appropriate

15. Line 167: the meaning of “linked to the site an error” is unclear. What kind of error?

16. Line 170: Eliminate “heavily” or give values.

17. Line 308: replace “phenotypic lag they persist” with “phenotypic lag that persists…”

18. Line 309-310: Remove question. It is unnecessary.

19. E.g. Line 331: Deng et al should be followed by a reference in each case it is mentioned.

20. Lines 425-426: Please re-write extended sentence of phenotypes generated.

21. Line 499: Define “various intervals”. Another way of looking at the very rare activities is as a selection for viability. Some of these events are very uncommon unless other frequent solutions are eliminated.

22. Line 504: Please state that no essential sequences are present telomere proximal to the break. This is obvious but would be helpful for the general reader.

23. Line 505: Maintain tense. Rewrite as” ….in fact be a feature of unstable chromosomes.”

24. Line511: Omit last sentence of the Discussion. The sentence ends the Discussion with an uncertain and not fully relevant issue.

25. Line 512: Paschini contribution can be thanked at this point or in acknowledgements.

26. Lines 545-547: Please rewrite. The description is difficult to read.

27. Line 556: Replace with: “…integration of the construct….”

28. Please do not use noc in the text. Write out word or define abbreviation.

29. Line 653: Replace “minimum” with “at least”

30. Line 657. The In-gel hybridization is mixed with the concurrent Southern blotting please. Please rewrite to clarify.

31. Were all plasmids obtained from the outside verified?

Reviewer #3: In this MS, Langston et al use a strain carrying an extra Chr12 VII with a CAN1 gene located at about 25 kb from the left telomere to study by genetic means the rearrangements of the extra chromosome. This system allows distinguishing several types of rearrangements caused by chromosomal replication errors in a given genetic context. They analysed these rearrangements at permissive or semi-permissive temperature in a mutant strain carrying the cdc13F648S ts allele. Importantly, recombinant ChrVII (Class 1) can be distiguished from unstable ChrVII or an outcome of the unstable ChrVII called class 2 recombinant.

The authors found that the cdc13F684S mutant exhibit an increased of the frequency of the three types of rearrangements in contrast of the stn1T223A,S250A mutant that behaves as the WT. Their genetic analyses based on the presence of genetic markers demonstrate that the Class 1 recombinant is probably linked to replication errors occurring in or near the telomere. In contrast, insertion of internal TG1-3 tracks at distant sites from the telomere (at which the G-rich strand is replicated by the lagging polymerase) does not generate instability. They further show that inactivation of Cdc13F684S in S phase (but not in G2/M) is at the origin of the instability and is associated with extensive ssDNA in telomeres. Interestingly, cells subjected to HU and then inactivated for Cdc13 exhibit a very high frequency of unstable chromosomes indicating a synergistic effect between Cdc13 inactivation and HU-induced depletion of dNTP. They further discovered that inactivation of Exo1, Sae2, and Pif1 suppress the formation of unstable chromosomes in favour of Class 1 recombinant. While the Class 1 recombinant generated in the cdc13F684S mutant is dependent on rad52Δ, the unstable chromosome does not depend on either Lig4 or Rad52 raising the question of the mechanism generating Class2 rearrangements.

This is a solid, smart, and interesting paper. There is not much to say on the data that are mainly rigorous genetic analyses. The overall conclusion that defective Cdc13 leads to a defect associated to the passage of the replication fork at or near telomeres further processed by Exo1 is important and of general interest. The fact that this defect is at the root of the simple or complex chromosomal rearrangements is also of general interest.

I propose some suggestions or slightly different interpretations that may improve the manuscript which is by essence not very easy to follow.

1) Could the authors make a scheme at the beginning of the MS for the broad readership of PloS Genetics explaining how the CAN marker could be lost.

2) The fact that the stn1T223A,S250A which has lost the ability to associate with Cdc13 does generate chromosomal rearrangements rather indicates that Stn1 is dispensable for the replication role of Cdc13. The phenotype of the double mutant may be explained by the fact that the Cdc13F684 could be less stable in the context of the stn1T223A,S250A

3) Explain what do you mean by replication-dependent chromosome capping

4) The fact that defective Cdc13 generates chromosome rearrangements at telomere and not at TG repeats inserted internally could be due to the rescue by another converging replication fork. This rescue would be only possible at internal TG repeats and not at telomeres.

5) Based on the absence of instability originating from the interstitial TG repeat, the authors propose that Cdc13 acts from the end of the chromosome rather than at the replisome. This argument is not maybe so strong (see point 4). In addition, Faure et al (Mol Cell 2010) showed that Cdc13 binds to the lagging telomere in a replication dependent way giving strength to the other hypothesis. The RPA like role of Cdc13 could explain the replication errors at or near the telomeres. This possibility should be at least discussed and the Faure Mol Cell paper cited.

6) Could the synergistic effect between Cdc13 inactivation and HU could be linked to the inhibition of late origin firing mediated by Mec1-Rad53? The synergistic effect somehow argues for a role of Cdc13 at the forks.

7) Hardy et al. Nature Comm (2014) showed that Sae2 and Sgs1 prevents accumulation of ssDNA at telomeres. This could be the simplest explanation to explain the role of Sae2. This paper should be cited.

8) I guess Class 1 rearrangements arise by Break Induced Replication, explaining the role of Pif1 in the generation of these rearrangements

9) Are Class 2 rearrangements depend on alternative-NHEJ?

10) The model is too complicated and should be simplified

**Have all data underlying the figures and results presented in the manuscript been provided?**

Reviewer #1: Yes

Reviewer #2: Yes

Reviewer #3: Yes

PLOS authors have the option to publish the peer review history of their article (what does this mean?). If published, this will include your full peer review and any attached files.

Reviewer #1: No

Reviewer #2: No

Reviewer #3: Yes: Vincent GELI

---

## [Decision Letter · Decision Letter 1]

23 Mar 2020

Dear Ted,

We are pleased to inform you that your manuscript entitled "Loss of Cdc13 causes genome instability by a deficiency in replication-dependent telomere capping" has been editorially accepted for publication in PLOS Genetics. Congratulations!

Yours sincerely,

Lorraine S. Symington

Associate Editor

PLOS Genetics

Gregory P. Copenhaver

Editor-in-Chief

PLOS Genetics

Comments from the reviewers (if applicable):

Reviewer's Responses to Questions

**Comments to the Authors:**

Reviewer #1: The authors have satisfactorily addressed all my concerns. Therefore I support the publication.

Reviewer #2: All my comments have been adequately answered.

Reviewer #3: The authors have answered to my concerns. I am satisfied with the revised version. The conclusions are important and supported by the data.

**Have all data underlying the figures and results presented in the manuscript been provided?**

Reviewer #1: Yes

Reviewer #2: Yes

Reviewer #3: Yes

PLOS authors have the option to publish the peer review history of their article (what does this mean?). If published, this will include your full peer review and any attached files.

Reviewer #1: No

Reviewer #2: No

Reviewer #3: No

**Data Deposition**

http://datadryad.org/submit?journalID=pgenetics&manu=PGENETICS-D-19-01631R1

**Press Queries**

---

## [Editor Report · Acceptance letter]

7 Apr 2020

PGENETICS-D-19-01631R1 

Loss of Cdc13 causes genome instability by a deficiency in replication-dependent telomere capping 

Dear Dr Weinert, 

We are pleased to inform you that your manuscript entitled "Loss of Cdc13 causes genome instability by a deficiency in replication-dependent telomere capping" has been formally accepted for publication in PLOS Genetics! Your manuscript is now with our production department and you will be notified of the publication date in due course.

With kind regards,

Jason Norris

PLOS Genetics

On behalf of:
